



# Comparison of wind-farm control strategies under realistic offshore wind conditions: wake quantities of interest

Kenneth Brown[1], Gopal Yalla[1], Lawrence Cheung[1], Joeri Frederik[2], Dan Houck[1], Nate deVelder[1], Eric Simley[2], and Paul Fleming[2]

[1]Wind Energy Technologies Department, Sandia National Laboratories, Albuquerque, NM 87123, USA
[2]National Wind Technology Center, National Renewable Energy Laboratory, Golden, CO 80401, USA

**Correspondence:** Kenneth Brown (kbrown1@sandia.gov)

**Abstract.** Wind-farm control strategies aim to increase the efficiency, and therefore lower the levelized cost of energy, of a wind farm. This is done by using turbine settings such as the yaw angle, blade pitch angles, or generator torque to manipulate the wake that negatively affects downstream turbines in the farm. Two inherently different wind-farm control methods have been identified in literature: wake steering (WS) and wake mixing (WM). As one of two companion papers focused on understanding practical aspects of these two wind-farm control strategies using large-eddy simulation (LES), we below analyze the wake quantities of interest for a single wind turbine performing WS and WM, while the companion article (Frederik et al., 2025) focuses on turbine quantities of interest including power and structural loads for the same computational setup and also includes two-turbine arrays with full and partial wake overlap. The simulations, which are based in the LES solver AMR-Wind, are tailored to have inflow conditions representative of measurements from a site off the east coast of the U.S. including with strong veer and low turbulence. The turbine, which is modeled in OpenFAST and coupled to the LES, is the IEA 15 MW, an open-source offshore design. After presenting an overview of the wake recovery for the different wake-control cases, the analysis probes the fluid-dynamic causes for the different performance of the arrays reported in the companion article by examining control volumes around the wakes and the budget of the mean-flow kinetic energy (MKE) within these volumes. In the high veer environment considered, the MKE recovery is dominated by mean convection, and this is shown to especially benefit the WS strategy when a neighboring turbine is directly downstream; there is $\approx 70\%$ more available power for a downstream turbine than the baseline case, and this power is gained primarily through mean convection on the left-tip and top-tip faces of the control volume. However, the case with imperfect knowledge of the exact wind direction favors the pulse-type WM strategy, largely because of $\approx 8\%$ increased turbulent entrainment from aloft versus the baseline that could be related to an apparent resistance to skewing in the pulsed wake. The general reduced effectiveness of helix-type and other individual-pitch-based WM strategies for inflow with high veer and low turbulence as reported in the companion paper is due, in part, to low magnitudes of phase-averaged turbulent entrainment. Two main findings of this study are thus that veer has a significant impact on the effectiveness of different wake-control strategies and that pulse-type WM may be a useful strategy when the objective is power maximization in realistic, offshore flow environments including imperfect knowledge of the exact wake overlap position on the downstream turbine.



# 1 Introduction

## 1.1 Overview of Wake Recovery

Measurements over a range of modern wind farms across Europe (Nygaard, 2014) and in the U.S. (El-Asha et al., 2017) show that the second row of turbines in a wind plant captures around 20% less than power than the first, and this number falls to around 40% for the turbines located deep within the array. In stable atmospheric conditions, losses up to 80% have been observed. The fundamental problem is the failure of the wake to fully "recover" from its depleted energy state after passing momentum to the turbine blades.

In large wind farm arrays, the contributions to wake recovery have been systematically examined by several authors using terms from the transport equation of MKE. LES (Calaf et al., 2010; Abkar and Porté-Agel, 2014) as well as measurements on lab-scale arrays of turbines (Cal et al., 2010; Hamilton et al., 2012) suggested that power extraction by turbines and power loss to turbulent production are primarily compensated by vertical flux of MKE from turbulent entrainment of the flow aloft. In the lower half of the wake, a smaller amount of MKE was fed into the flow below the rotor layer through turbulent entrainment (Cal et al., 2010; Calaf et al., 2010).

As with the larger wind-farm studies above, several authors have performed detailed MKE budget analyses on the wakes of isolated turbines. Lebron et al. (2012), who measured a scaled turbine, produced budget measurements on a streamtube control volume and found that radial turbulent transport is most responsible for re-energizing the wake. Boudreau and Dumas (2017) performed a related analysis using simulations with the unsteady Reynolds-averaged Navier-Stokes equations to show again that turbulent transport is the dominant contributor to wake recovery for all locations downstream of the tip vortex breakdown. Their analysis also reveals a secondary, but relevant, term contributing to the wake recovery, which is the mean radial velocity. Indeed, it is the turbulent transport and the mean convection that are targeted by the wake-control techniques to be reviewed next.

## 1.2 Wake-Control Techniques

In the last 10 years, strategies to combat wake effects have emerged in the form of wake-control techniques. These techniques see upstream turbines operating at sub-optimal (i.e., non-greedy) set-points for the benefit of downstream turbines. Wake-control techniques broadly fall into three categories: wake reduction (i.e., turbine derating), wake steering, and wake mixing. Each technique has shown potential to produce downwind flow with higher velocities. While the wake-recovery mechanisms of the wake-reduction technique are the same as those of conventionally-operated turbines as reviewed above, WS and WM, which are illustrated in Fig. 1, use additional mechanisms.

Wake steering, which involves the intentional misalignment of the yaw of a turbine to the incoming wind direction, takes advantage of the conservation of momentum in the spanwise direction, which dictates that the fluid deflect from the streamwise direction due to the spanwise component of force exerted by the yawed turbine on the flow (Jiménez et al., 2010; Gebraad et al., 2016). Intelligent application of yaw misalignment on upstream turbines can therefore deflect, or steer, a wake away from a downstream turbine, thus resulting in higher flow velocities and larger power production for nearby, downwind turbines



(Fleming et al., 2019; Howland et al., 2019). Another consequence of the spanwise deflection is the formation of a counter-rotating vortex pair (CVP), positioned in the wake such that a stream-normal cross section of flow behind a moderately or strongly yawed turbine will yield a kidney-bean shape due to the vorticity field of the CVP (Bastankhah and Porté-Agel, 2016). A mechanism for improved wake recovery is also present, which is the replacement of lower-momentum wake flow with higher-momentum ambient flow aloft according to the circulation of the upper vortex of the CVP, and secondary steering of downstream wakes also occurs (King et al., 2021). In the case of positive yaw misalignment (where positive is defined as a counter-clockwise offset of the yaw heading from the inflow wind vector when viewed from above), the swirl of the wake combines constructively with the upper vortex of the CVP, and the wake center is deflected both upwards and away from the turbine centerline (Bastankhah and Porté-Agel, 2016). One challenge in the implementation of wake steering is its sensitivity to turbulent variations in wind direction, but strategies for optimization under uncertainty (OUU) of the wind direction have been developed and may increase the power production significantly versus an optimal solution that considers only the time-averaged wind direction (Simley et al., 2020).

Wake mixing, which may involve periodic variations of blade pitch, rotor torque, or yaw, actuates flow structures to achieve improved wake recovery. The structures may be tip vortices (Marten et al., 2020; Brown et al., 2022) or, possibly more commonly, larger-scale coherent structures (Munters and Meyers, 2018a; Frederik et al., 2020a). For these latter structures, actuation is often done through collective pitch (i.e., pulse method) or individual pitch (i.e., helix method and others). As the effectiveness of the pulse and helix-related techniques is sensitive to the Strouhal number of the periodic variations, the WM mechanism has been hypothesized to be related to normal modes of the wake (Cheung et al., 2024). The vortical structures of these normal modes, which have been shown to naturally exist in wind turbine wakes at smaller magnitudes without intentional forcing (Okulov et al., 2014), are shed periodically from the rotor disk to produce added turbulent entrainment through higher magnitudes of kinematic shear-stress around the wake edges (Munters and Meyers, 2018b; Cheung et al., 2024). Yalla et al. (2025a) argue along these lines by quantifying amplitudes of the modes excited by the different WM strategies and drawing comparison between these modes and increased turbulent entrainment of MKE in the actuated wakes. For the helix method particularly, Korb et al. (2023) noted the increase in turbulent entrainment and also suggested that helically-winding deflection of the wake makes a significant contribution to the available energy for a downstream turbine (i.e., by deflecting or spreading the wake deficit over a wider cross-stream area). Van der Hoek et al. (2024) apply phase averaging to draw attention to the effect of the helix approach on the breakdown of the near-wake tip-vortex structure.

While the above studies demonstrate the ability of wake-control strategies, namely WS and WM, to effect positive change on the wind fields experienced by nearby turbines, the performance of these strategies has only been compared side-to-side in limited instances (Frederik et al., 2020b; Coquelet et al., 2022; Taschner et al., 2024) and never with inflow conditions that are representative of offshore, stable atmospheric boundary layers (ABLs). Such shortcomings limit researchers' understanding of the utility of WS and WM as well as constrain the development of practical implementations of such strategies. Our companion paper series addresses this gap by comparing WS and WM in realistic, offshore inflows including significant veering of the flow. The companion article (Frederik et al., 2025) quantifies differences between the wake-control strategies in terms of the turbine quantities of interest of power and structural loads while this article focuses on deeper investigation of the underlying





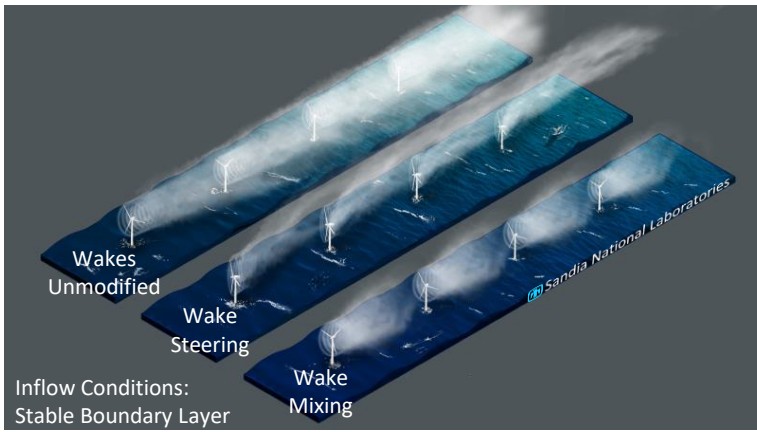

**Figure 1.** Depiction of wake effects and two wind-farm control strategies.

fluid-dynamic mechanisms responsible for the differences in wake effects. To this end, we herein quantify for the first time the relative importance of the terms in the full budget of MKE during wake recovery for WS and WM strategies as compared to the case of a conventionally-operated wind turbine.

The article begins by introducing the control-volume analysis, LES domain, inflow conditions, sampling planes, turbine model, and controller setup in Sect. 2. The results presented in Sect. 3 first detail the higher-level findings of the control-volume study before analyzing the phenomena contributing to the dominant recovery terms. Conclusions are drawn in Sect. 4.

## 2 Methodology

### 2.1 Control-Volume Analysis

A control-volume analysis is useful to uncover the source of differing wake recovery rates between the wake-control strategies. We begin with Lebron et al. (2012)'s transport equation for axial mean-flow kinetic energy per unit density, $\frac{1}{2}\overline{u}^2$, derived from the steady Reynolds-Averaged Navier–Stokes (RANS) equation for a three-dimensional high-Reynolds-number flow in the axial direction and neglecting viscosity due to absence of nearby wall boundaries, Eq. (1),

$$\nabla \cdot \left( \overline{\mathbf{u}} \frac{1}{2} \overline{u}^2 \right) = -\frac{\overline{u}}{\rho}\frac{\partial \overline{p}}{\partial x} + \nabla \cdot (\overline{u}\mathbf{T}_x) - \mathcal{P}_x + \overline{u}f_b, \tag{1}$$

where $\rho$ is density, $\overline{p}$ is mean pressure, $\overline{\mathbf{u}} = (\overline{u}, \overline{v}, \overline{w})$ is the mean velocity vector, $\mathbf{T}_x = (-\overline{u'u'}, -\overline{u'v'}, -\overline{u'w'})$ are the $x$-components of the Reynolds stress tensor, $\mathcal{P}_x = -\overline{u'u'}\frac{\partial \overline{u}}{\partial x} - \overline{u'v'}\frac{\partial \overline{u}}{\partial y} - \overline{u'w'}\frac{\partial \overline{u}}{\partial z}$ is the production of turbulence kinetic energy (i.e., a sink for the mean-flow kinetic energy) from the axial portion of mean kinetic energy neglecting the contributions to





production from the lateral portions, $f_b$ are body forces within the domain, and the dissipation due to the mean flow has been
neglected due to the high Reynolds number (Pope, 2001).

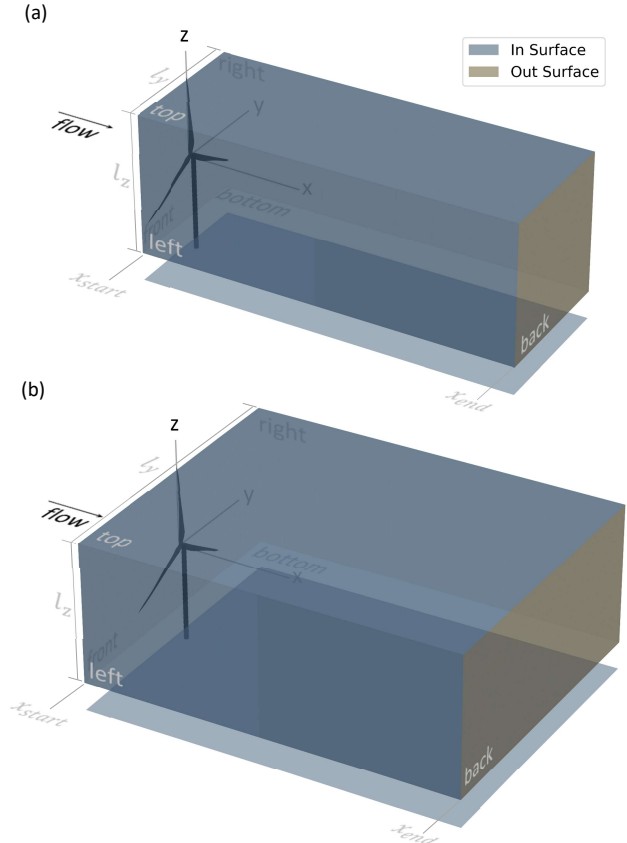

**Figure 2.** Depiction of the control-volume faces used to analyze the budget of mean-flow axial kinetic energy for the control volumes with
(a) $l_y = \frac{1}{2}\sqrt{\pi}D$ ($\approx 0.9D$) and (b) $l_y = (\frac{1}{2}\sqrt{\pi}+1)D$ (1.9D). Both cases have $l_z = \frac{1}{2}\sqrt{\pi}D$ ($\approx 0.9D$). The turbine sits just ahead of the front
face, which is at $x_{start} = 0.1D$, and is not included in the volume. The volumes are positioned to capture the wake from an upstream turbine
as well as the inflow region for a hypothetical downstream turbine with varying levels of wake overlap.

Eq. (1) can be integrated over a volume such as those shown in Fig. 2, and, after conversion of the two resulting volume
integrals with divergence operators to surface integrals using the Gauss theorem, we have Eq. (2),

$$\iint \frac{1}{2}\overline{u}^2\overline{\mathbf{u}} \cdot \mathbf{n} \, dS = \iint \overline{u}\mathbf{T_x} \cdot \mathbf{n} \, dS - \frac{1}{\rho}\iiint \overline{u}\frac{\partial \overline{p}}{\partial x} \, dV - \iiint \mathcal{P}_x \, dV + \iiint \overline{u}f_b \, dV \qquad (2)$$

The lateral limits of integration are specified as seen in Fig. 2 by $l_y$ and $l_z$, which are symmetric about the planes of $y = 0$ and
$z = 0$, respectively. In an effort to compensate for the use of a square-shaped cross section rather than a circular one, one case





that will be considered below is that of $l_y = l_z = \frac{1}{2}\sqrt{\pi}D$ (i.e., Fig. 2(a)), which are the dimensions of the square with an area equal to that of a circle of diameter $D$. The streamwise limits of integration are $x_{start}$ and $x_{end}$, and $x_{start}$ will always be taken to be $0.1D$ while $x_{end}$ will be variable. Note that, although the volumetric integral of $\overline{u}f_b$ will have several contributions as will be itemized further below, there is no contribution from the turbine since the blades and tower reside outside these limits
of integration as depicted in Fig. 2.

We next expand the left-hand side of the equation to correspond to different surfaces around the volume as in Eq. (3),

$$\iint \frac{1}{2}\overline{u}^2\overline{\mathbf{u}} \cdot \mathbf{n}\, dS = -\iint_{fr} \frac{1}{2}\overline{u}^3\, dS \pm \iint_{lf,rg} \frac{1}{2}\overline{u}^2\overline{v}\, dS \pm \iint_{tp,bt} \frac{1}{2}\overline{u}^2\overline{w}\, dS + \iint_{bk} \frac{1}{2}\overline{u}^3\, dS. \tag{3}$$

where the subscripts of the integrals are abbreviations of the surfaces denoted in Fig. 2 (i.e., $fr$ = front, $lf$ = left, $rg$ = right, $tp$ = top, and $bt$ = bottom) and where the signs before the integrals of the second and third terms on the right-hand side are
opposite for the two limits of integration in each term. Note that, whereas Lebron et al. (2012) applied a control volume over a streamtube of the wind turbine and thus reduced the surface integrals to contributions from only the inlet and outlet planes, we desire a control volume that has fixed location and dimensions relative to a potential downstream turbine rather than being dependent on the wake trajectory. Thus, all terms in the surface integral along the streamwise length of the volume as shown in Eq. (3) must be retained.

The first term on the right-hand side of Eq. (2) can similarly be expanded as Eq. (4),

$$\iint \overline{u}\mathbf{T_x} \cdot \mathbf{n}\, dS = \iint_{fr} \overline{u}(\overline{u'u'})\, dS \pm \iint_{lf,rg} \overline{u}(\overline{u'v'})\, dS \pm \iint_{tp,bt} \overline{u}(\overline{u'w'})\, dS - \iint_{bk} \overline{u}(\overline{u'u'})\, dS. \tag{4}$$

Eq. (2) represents a budget of density-normalized MKE through a control volume of interest to a wind farm's performance. Particularly, it is the $bk$ terms in Eq. (3) and Eq. (4) that govern the flux, $AP_{out}$, that flows out of the back plane of the volume and is available for a potential downstream turbine to hopefully harvest as in Eq. (5),

$$AP_{out} = \underbrace{\iint_{bk} \frac{1}{2}\overline{u}^3\, dS}_{\phi_{mean}} + \underbrace{\iint_{bk} \overline{u}(\overline{u'u'})\, dS}_{\phi_{turb}} \tag{5}$$


where the $\phi_{mean}$ and $\phi_{turb}$ terms are the mean and turbulent fluxes, respectively, of axial MKE at the back face of the control volume. Collecting all other terms, which represent the balance of MKE changes due to the inflow and side faces, as well as due to processes within the control volume, on the opposite side of the equation yields $AP_{in}$ as in Eq. 6,



$$
AP_{in} = \underbrace{\iint\limits_{fr} \frac{1}{2}\overline{u}^3 \, dS \pm \iint\limits_{lf,rg} \frac{1}{2}\overline{u}^2\overline{v} \, dS \pm \iint\limits_{tp,bt} \frac{1}{2}\overline{u}^2\overline{w} \, dS}_{\phi_{mean}} + \underbrace{\iint\limits_{fr} \overline{u}(\overline{u'u'}) \, dS \pm \iint\limits_{lf,rg} \overline{u}(\overline{u'v'}) \, dS \pm \iint\limits_{tp,bt} \overline{u}(\overline{u'w'}) \, dS}_{\phi_{turb}} -
$$

$$
\underbrace{\iiint \mathcal{P}_x \, dV}_{\mathcal{P}} - \underbrace{\frac{1}{\rho}\iiint \overline{u}\frac{\partial \overline{p}}{\partial x} \, dV}_{\mathcal{W}_p} + \underbrace{\iiint \overline{u}f_b \, dV}_{\mathcal{W}_{bf}}
\tag{6}
$$

where the $\phi_{mean}$ terms are the fluxes of axial MKE into or out of the control volume due to mean convection, $\phi_{turb}$ terms are the likewise fluxes due to turbulent entrainment, the $\mathcal{P}$ term is the loss of axial MKE due to production of turbulence kinetic energy, the $\mathcal{W}_p$ term is the change in axial MKE due to pressure work, and $\mathcal{W}_{bf}$ is the change in axial MKE due to body forces.

In the results to follow, we will initially quantify $AP_{out}$ for different cases of interest and compare these values to $AP_{in}$ to verify that certain approximations in the derivation as described above do not significantly influence the balance of the MKE

budget. Then, a detailed tracking of the $AP_{in}$ terms will inform which terms contribute the most to the replenishment of $AP_{out}$ in the wake of a wind turbine employing different wake-control strategies.

## 2.2   LES Domain

The simulations in this work were performed with the U.S. Department of Energy (DOE) ExaWind solver AMR-Wind (Sprague et al., 2020; Sharma et al., 2024), a massively parallel, block-structured adaptive-mesh, incompressible flow solver for wind tur-

bine and wind farm simulations. The AMR-Wind solver uses a second-order finite-volume method with second-order temporal integration, based on the approximate projection method of Almgren et al. (1998) and Sverdrup et al. (2018). For the solution of atmospheric boundary layers (ABLs) and wind-farm physics, AMR-Wind includes the following body force terms: turbine actuator forcing, Boussinesq buoyancy, Coriolis forcing, and a body force to maintain the precursor-derived inflow condition in the presence of the turbine's blockage. In the control-volume analysis to follow, the first two terms above were ignored since

the turbine is not included in the control volume and the buoyancy acts primarily in the vertical direction, respectively. The second two terms were accounted for through the $\mathcal{W}_{bf}$ term of Eq. (6), using the latitude of the simulation and a prescribed body force, respectively.

Simulating wind turbines within a turbulent ABL was a two-step process in AMR-Wind. First, a desired ABL condition was established using precursor simulations. The boundary layer was initialized with small velocity and temperature perturbations

near the surface to accelerate turbulence development, and the precursors were run for tens of thousands of seconds to establish fully developed turbulent flow. The first and most important simulation domain used in this article was 4560 m by 2000 m by 960 m in the x-, y-, and z-directions, respectively, using the same coordinate system as Fig. 2. This domain was found suitable for a stable ABL condition while a second domain of dimensions 7200 m by 4000 m by 1440 m in the x-, y-, and z-directions, respectively, was utilized for a near-neutral ABL condition with larger atmospheric structures. The smaller and

larger precursors used isotropic, uniform mesh sizes of 5 m and 10 m, respectively.



**Table 1.** Specifications of the IEA 15 MW reference turbine (Gaertner et al., 2020). Values labeled "design" refer to region 2 operation.

| Name | Value |
|---|---|
| Hub-height | 150 m |
| Rotor diameter | 240 m |
| Rated wind speed | 10.59 m/s |
| Design Ct | 0.804 |
| Design TSR | 9.0 |

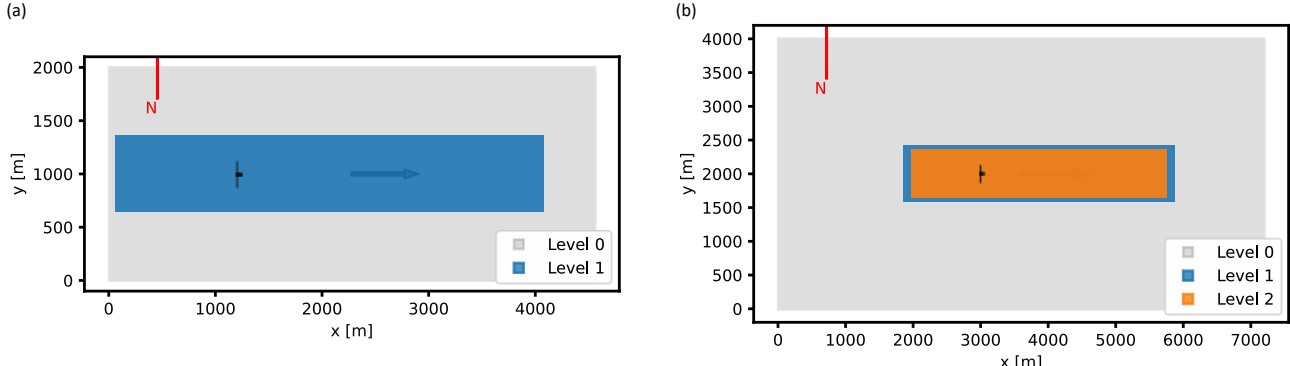

**Figure 3.** Top-down view of LES domains for the (a) MSLT and (b) MSMT wind conditions. The most refined level of each mesh is 2.5 m isotropic.

Second, the turbine was introduced into AMR-Wind through coupling with the OpenFAST software suite (National Renewable Energy Laboratory, 2024b), which has been the subject of validation work for the case of turbines without coupling to LES (Brown et al., 2024) and with coupling to LES (Hsieh et al., 2024). The OpenFAST model was the open-source IEA 15 MW reference turbine, which has the general specifications shown in Table 1; full details for this model are available in Gaertner et al. (2020) and omitted here for brevity. An actuator line model (ALM) is used to spread the turbine aerodynamics forces computed in OpenFAST to body forces in the LES that are distributed to the surrounding fluid (Sorensen and Shen, 2002). The ALM is defined with an isotropic Gaussian projection function with spreading parameter $\varepsilon/\Delta x = 0.8$. The turbine was located on the domain centerline and surrounded by a refinement region with 2.5 m resolution for both of the LES domains considered in this study and pictured in Fig. 3. It is worth mentioning that energy is not strictly conserved at the transitions between refinement regions. In this work, our region of interest extends beyond the most refined region for the larger domain in Fig. 3(b), however, any error from non-conservation of energy outside this region appears small. The turbine was simulated for 1800 seconds using a time step of 0.02 seconds. The first 560 s were discarded as transient startup, leaving 1240 s, or 14 complete Strouhal cycles to be described below, for analysis.





### 2.3 Inflow Conditions

The inflow-condition targets for the precursors described above were derived from a floating-lidar measurement campaign performed by DNV (Mason, 2022). Specifically, data were drawn from 1.6 years of measurements at the E06 proposed siting area in the New York Bight. E06 was chosen since it was the larger of the two siting areas in the measurement campaign and had a coastal standoff more representative of several other potential wind farms in the area. Measurements were performed with a ZephIR ZX300M lidar and an EOLOS FLS-200 Buoy system, which underwent two pre-validations, one onshore and

one offshore, before deployment, and measurement uncertainty of wind speed was estimated at 3.3%. The atmosphere was sampled at heights of 20 to 200 m in increments of 20 m and reported at 10-minute intervals. All data is publicly available (DNV, 2023).

Mason (2022) reports that the corrected turbulence intensity (TI) from the lidars was assessed to be unrealistically high. Therefore, the present authors decided to recalculate TI simply as the standard deviation of wind speed divided by the mean

wind speed for each measurement height. This approach introduces a known effect of underestimating TI due to negligence of the lidar's probe-volume averaging, though there exists a competing effect of over-estimation due to cross-contamination of different velocity components for surface-based lidars (Sathe et al., 2015).

Several pre-processing steps were applied to the 10-minute data. Hub-height values of wind speed and TI were linearly interpolated from the measurement heights. Values of shear exponent and veer were calculated with a power-law fit of the wind

speed and a linear fit of the wind direction, respectively. Note that the maximum height of the measurement was 200 m, which is 70 m below the top-tip location for the IEA 15 MW turbine to be studied, so the number of degrees of veer over the rotor height was calculated by extrapolating the linear fit to the top of the assumed rotor.

Filtering operations were next implemented. Since the strength of wake effects are sensitive to wind speed and TI, the 10-minute data were filtered into three wind-speed bins and three TI bins according to Table 2. To avoid compromising the

dominant mean atmospheric behavior with a small number of occurrences of low-level jets, only bins with approximately power-law wind-speed profiles were retained in the binning of Table 2, and the criterion for acceptance was a coefficient of determination greater than 0.5 on the power-law fit. The last column in Table 2 indicates that the percent of remaining data after power-law filtering was always greater than 77%. The mean hub-height wind speed, hub-height TI, rotor shear coefficient, and rotor veer were calculated over the bins in each of the conditions in Table 2.

From the nine wind conditions identified in Table 2, two will be considered in this article. Specifically, the low- and medium-TI cases for the medium wind speed (MSLT and MSMT, respectively) were selected since the longevity of wakes in low and medium turbulence conditions motivates the use of wake-control technology. Notably, the MSLT and MSMT cases account for more than 73% of all the measured wind instances at the medium wind speed, which is just below the rated wind speed of existing offshore turbines, and thus the relevance of the MSLT and MSMT cases to the performance of a wind farm is significant

since the rated condition sees high energy production and still significant wake effects. Wind conditions from several of the other cells from Table 2 are also considered in Frederik et al. (2025).





**Table 2.** Frequency of occurrence for different wind conditions from the measured data. The percentage values refer to the percent of data within each wind-speed range for a given TI level, and the values in parenthesis are the corresponding number of 10-minute bins. The combined sum over a row does not add to 100% because of the filtering of some cases with poor power-law fits as described in the text. Conditions in bold are those considered in this article.

|  | Low TI ($\leq$5%) | Med. TI (5-10%) | High TI ($\geq$10%) | Combined TI |
|---|---|---|---|---|
| Low wind speed (6-7 m/s) | 24.3% (1856 bins) | 35.1% (2676 bins) | 18.5% (1414 bins) | 77.9% (5946 bins) |
| Med. wind speed (8.5-9.5 m/s) | **30.5% (2325 bins)** | **43.3% (3298 bins)** | 13.0% (989 bins) | 86.8% (6612 bins) |
| High wind speed (11-12 m/s) | 30.3% (1910 bins) | 53.5% (3371 bins) | 9.7% (614 bins) | 93.5% (5895 bins) |

After the measurement data was processed as above, precursor data was generated for the MSLT and MSMT conditions using surface roughnesses of 0.0005 and 0.0013 m, respectively, surface temperature rates of -0.12 K/hr and 0 K/hr, and surface temperature fluxes of 0 K-m/s and 0.001 K-m/s. A process was completed to downselect specific windows of interest from the 10000s of precursor data to match the measurement targets. The windows were selected based on the agreement of the four quantities already discussed (i.e., mean hub-height wind speed, hub-height TI, rotor shear coefficient, and rotor veer) as well as the hub-height wind-direction error. This last metric is the deviation in hub-height wind direction from the nominal streamwise (i.e., $x$) direction for the simulation window selected. This deviation is a consequence of stochastic fluctuations in the simulated atmosphere and should be borne in mind when comparing wake-control results since the control volume positions as well as the turbine arrays in the companion article will be positioned according to the Cartesian coordinate system rather than the hub-height wind direction.

The achieved values for the selected windows are presented in Table 3 alongside the targets from the measurement. The order of importance of the criteria used for selecting the periods was (from highest to lowest): wind speed, TI, shear, veer, and hub-height deviation in wind direction. While the first three have been generally well matched, the strong veer magnitude measured at the NY Bight proved difficult to recreate exactly in LES. However, the veer magnitudes achieved in the simulations represent a significant increase in magnitude compared to most previous wake-control studies. For the MSLT case that is the focus herein, the error in hub-height wind direction amounts to a deviation of less than 1.3% rotor diameter for the IEA 15 MW at a typical row-to-row spacing of five diameters. For the MSMT case, the value is 2.0%.

Comparison of mean vertical profiles between the simulated data and the average measured data for the two cases of Table 3 is given in Figure 4. A noticeable shortcoming of the measurement dataset is the maximum height of the measurement at 200 m, which is 70 m below the top-tip location for the IEA 15 MW. Given this limitation, the profiles of wind speed and TI show good agreement with the measurement. As noted above, the simulated wind-direction profiles of both wind conditions indicate an under-realization of veer compared to the measurement.





**Table 3.** Mean simulated values of inflow for the two wind conditions considered herein. The values in parentheses are the the corresponding mean values from the measurement bins. The simulated air density was taken as 1.2456 kg m$^{-3}$, which was the mean air density over the measurement dataset. Note the values in this table do not identically match those reported in Frederik et al. (2025) because the time window of interest was 550-750 s longer in the present article to improve convergence of second-order flow statistics.

| Description | Measurement Filters | | Simulated (Measured) Values | | | | |
|---|---|---|---|---|---|---|---|
| | Hub-height wind speed [m s$^{-1}$] | Hub-height TI [-] | Hub-height wind speed [m s$^{-1}$] | Hub-height TI [-] | Rotor-avg. shear [-] | Rotor-avg. veer [$^o$] | Hub-height WD err. [$^o$] |
| Med. wind speed, low TI (MSLT) | [8.5,9.5] | [0,0.05] | 9.01 (9.03) | 0.0309 (0.0371) | 0.160 (0.171) | 8.94 (21.3) | 0.14 |
| Med. wind speed, med. TI (MSMT) | [8.5,9.5] | [0.05,0.10] | 9.03 (9.01) | 0.0700 (0.0698) | 0.0655 (0.0677) | 1.1 (5.8) | 0.23 |

## 2.4 Sampling Planes

235 The sampling planes are depicted in Fig. 5 and include $xy$ and $xz$ planes for calculation of the surface fluxes on the side planes as well as $yz$ planes for calculation of both the surface fluxes on the front and back planes and for estimation of the volumetric integrals. All planes are sampled at a resolution of $0.025D$ (6 m) and a frequency of 2 Hz. Considering the evaluation of Eqs. (5) and (6), the surface integrals benefit from the relatively fine resolution of $0.025D$ on $xy$ and $xz$ planes, if available, while the volumetric integrals suffer from relatively coarse resolution of $0.15D$ to $1D$ for the $yz$ planes in the $x$-direction due 240 to data storage limitations. As these volumetric terms are not expected to be dominant in the MKE budget (Lebron et al., 2012), this uncertainty is accepted. To facilitate post-processing, these $yz$ planes are linearly interpolated in the streamwise direction to produce the same $0.025D$ streamwise resolution as the $xy$ and $xz$ planes. Derivatives of the raw velocity and pressure for all the data are then taken using using second-order-accurate central differences in the interior points and either first- or second-order-accurate one-sided differences at the boundaries. For some of the control volume analyses, the proper $xy$ and $xz$ planes 245 were not available, and these cases therefore relied on the streamwise-interpolated $yz$ planes. Error quantification is provided further below.

## 2.5 Controller Setup

To implement the different control strategies on the IEA 15 MW turbine, NREL's reference open-source controller (ROSCO v2.8.0; National Renewable Energy Laboratory (2024a)) is used. ROSCO was developed to offer the scientific community a 250 baseline wind turbine reference controller with industry-standard functionality.



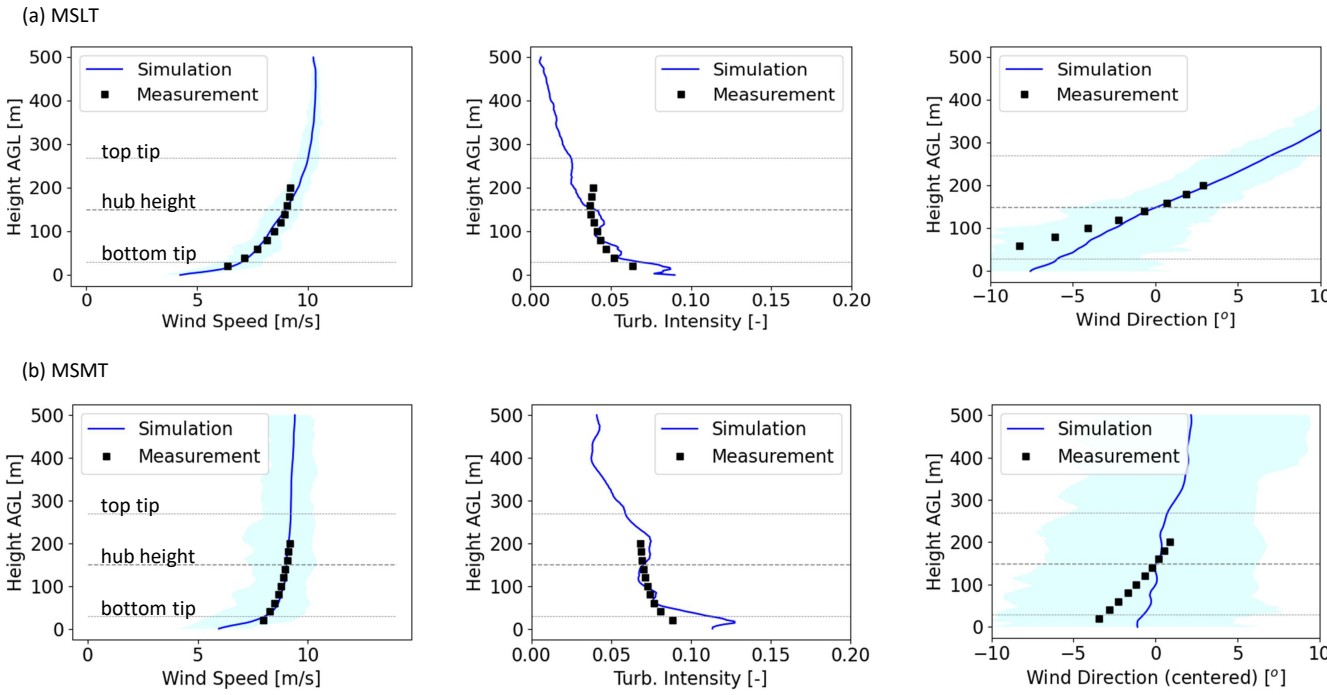

**Figure 4.** Comparison of vertical profiles between the simulated data for the (a) MSLT and (b) MSMT wind conditions as described in Table 3. The light blue shading indicates the range of the (first-order) statistics from the simulation data. Before averaging the wind-direction profiles from the measurement, the hub-height wind direction was subtracted from each bin's data.

For the work executed in this paper, WM functionality had to be added to ROSCO. This functionality was included in ROSCO version 2.8.0 including both a normal-mode and Coleman-transform method. In this article, we choose to use the normal-mode representation of WM rather than the Coleman-transform representation, although both methods produce equivalent results as demonstrated in Cheung et al. (2024). Following the derivation in Cheung et al. (2024), we arrive at the time series of pitch amplitude for any blade, $\theta_b(t)$, as in Eq. 7

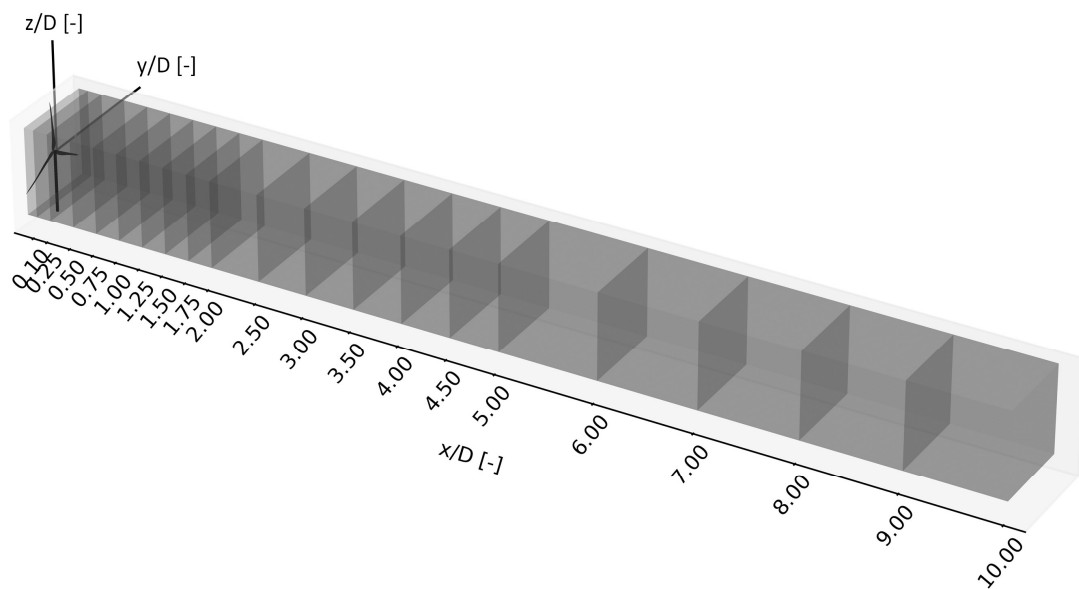

**Figure 5.** Sampling planes used in this study including $xy$, $xz$, and $yz$ planes. The planes shown are cropped according to the dimensions of the smaller control volume.

$$\theta_b(t) = \theta_0 + A \sum_j \cos\big(\omega_e t - n_j(\psi_b(t) + cl)\big). \tag{7}$$

where $\theta_0$ is the nominal blade-pitch command, $A$ is the pitch amplitude of each mode to be forced, $\omega_e$ is the angular frequency according to the Strouhal number, $St$ (i.e., $\omega_e = 2\pi St U_{hh}/D$ where $U_{hh}$ is hub-height wind speed), $n_j$ is the azimuthal mode number for mode $j$, $\psi_b$ is the time series of blade azimuthal angle, and $cl$ is the azimuthal clocking angle constant. Input values for Eq. (7) that will be examined in this article are shown below in Table 4. Notably, the WM cases include the pulse or $n = 0$, counter-clockwise helix (looking downstream) or $n = -1$, side-side or $n = \pm 1, cl = 90°$, and up-down or $n = \pm 1, cl = 0^o$.





Note that the counter-clockwise helix has been found to be more promising for inducing wake recovery than the clockwise helix (Frederik et al., 2020a).

**Table 4.** Control settings for the WM strategies as implemented in ROSCO. The values for the Coleman method are shown for reference while the values for the normal-mode method correspond to the inputs to Eq. 7.

| | General settings | | Coleman method | | | Normal-mode method | |
|---|---|---|---|---|---|---|---|
| **Control strategy** | **A** | $St$ | **n** | $\phi_{\text{tilt}}$ | $\phi_{\text{yaw}}$ | **n** | **cl** |
| $n = 0$ (pulse) | 4 | 0.3 | 0 | $0°$ | n/a | 0 | $90°$ |
| $n = -1$ (ccw helix) | 4 | 0.3 | 1 | $0°$ | $90°$ | $-1$ | $90°$ |
| $n = \pm 1, cl = 90^o$ (side-to-side) | 2 | 0.3 | 1 | n/a | $0°$ | $+1, -1$ | $90°$ |
| $n = \pm 1, cl = 0^o$ (up-and-down) | 2 | 0.3 | 1 | $0°$ | n/a | $+1, -1$ | $0°$ |

Finally, for the WS case, no WM control is implemented. For this case, the turbine runs using the same control settings as
the baseline case with the exception of a constant $+20°$ yaw misalignment of the rotor with respect to the $-x$ direction. The positive yawing direction is defined as counterclockwise when seen from above.

## 3  Results

The analysis below first provides a high-level survey of the wake behavior of the different control strategies in the MSLT condition. Next we present the findings of the control-volume study before analyzing in greater depth the phenomena contributing
to the dominant recovery terms. Results will finally be compared and contrasted with those of the MSMT condition. All results are from the MSLT condition unless otherwise noted.

### 3.1  Survey of the wake recovery

We first consider a qualitative perspective of the wake recovery based on cross-sections of the wake flow shown in Figure 6. For all cases, the wake is observed to create an initially sharp shear layer around the circle traced by the rotor tips, and a hub jet and
tower shadow are also apparent. Moving downstream, the sharp lateral gradients give way to an increasingly homogenized flow field as wake recovery processes take hold. A hallmark feature of the far wakes is the strong wake skewing due to the presence of veer in the inflow, and a more detailed analysis of the wake skewing is executed in the companion paper (Frederik et al., 2025). We can also observe differences in wake shapes between the different control strategies. WS is shown here to deflect the wake towards negative $y$ values and introduce a kink that produces the characteristic kidney-bean shape often produced in
steered wakes by the CVP as discussed above. As expected, WS control also slightly narrows the wake, which is in contrast to the WM strategies that produce a widening effect. The $n = 0$ strategy in Figure 6(c) is an outlier among the WM cases and sees reduced wake-skewing.

A more quantitative perspective on the wake recovery is afforded by Figure 7, which plots the streamwise development of MKE recovery using the near-wake-subtracted and normalized $AP_{out}$, distinguishing the turbulent contribution to $AP_{out}$ from

eawe
european academy of wind energy

WIND
ENERGY
SCIENCE
DISCUSSIONS



**Figure 6.** Cross-sections of normalized $\overline{u}$ at five $x/D$ locations in the wakes for each control strategy. The black boxes are centered on the turbine hub position and have side length of $\frac{1}{2}\sqrt{\pi}D$ ($\approx 0.9D$). The flow is viewed from upstream looking downstream.

the mean one since turbulence of relevant lengthscales has certain effects on the power curve of downstream turbines (Wagner et al., 2010). Figure 7(a) shows results for cross-sectional side lengths of $l_y = l_z = \frac{1}{2}\sqrt{\pi}D$ ($\approx 0.9D$), and here the efficacy of WS to recover $AP_{out}$ is better than that of the WM strategies for $x > 3.5D$. This result changes, however, when the horizontal side length $l_y$ is increased to $(\frac{1}{2}\sqrt{\pi}+1)D$ ($\approx 1.9D$) in Figure 7(b), where the WM strategy of $n = 0$ shows strongest wake recovery from $x = 2.5D$ to $9D$. While the narrower spanwise dimension of $l_y = \frac{1}{2}\sqrt{\pi}D$ is relevant to downstream turbines if

the wind direction is exactly aligned with a wind turbine column, $l_y = (\frac{1}{2}\sqrt{\pi}+1)D$ has implications for downstream turbines in a wider range of wind directions. The $l_y = (\frac{1}{2}\sqrt{\pi}+1)D$ value corresponds to an uncertainty in the lateral offset of the



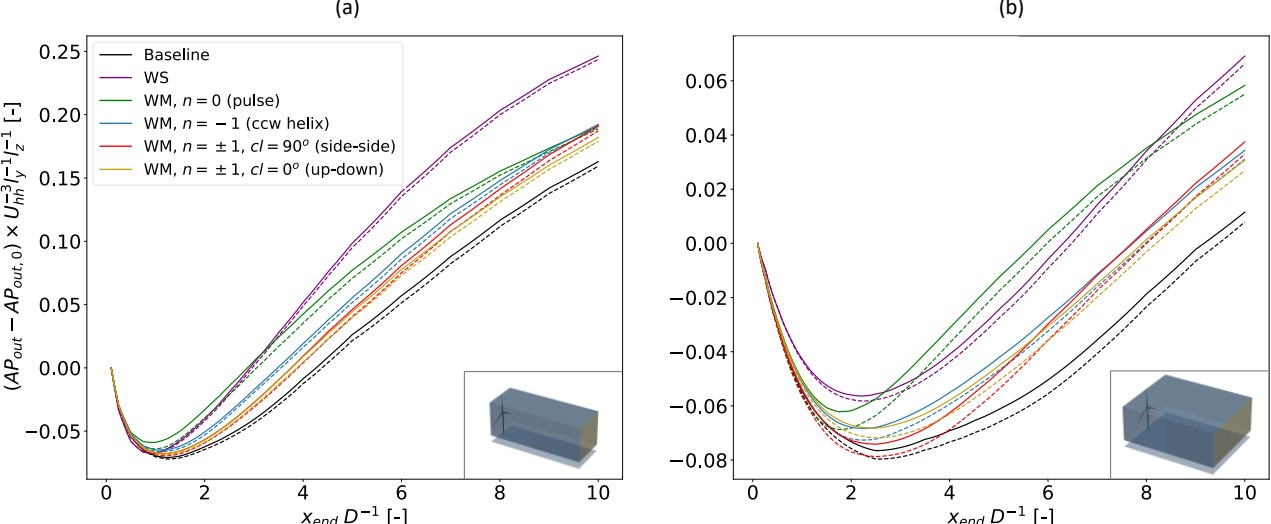

**Figure 7.** Streamwise development of MKE flux at the back surface, $AP_{out}$, from the control volume for (a) $l_y = l_z = \frac{1}{2}\sqrt{\pi}D$ and (b) $l_y = (\frac{1}{2}\sqrt{\pi} + 1)D$ and $l_z = \frac{1}{2}\sqrt{\pi}D$. Solid lines are the sum of both mean and turbulent terms of Eq. 5 while dashed lines are the mean term only. $AP_{out,0}$ refers to $AP_{out}$ at $x_{start}$ (i.e., $x = 0.1D$).

downstream turbine of $\pm 0.5D$, or $\approx \pm 5°$ variation in viewing angle from the upstream turbine for a $5D$ turbine spacing, and was motivated by the $\pm 5°$ range of wind directions considered in Taschner et al. (2024) to be the range of variation over which a turbine yaw controller does not yet react to small variations in wind direction (c.f., the yaw error standard deviation

of $5.25°$ in Simley et al. (2020)). Similar to Taschner et al. (2024), we find that the WM strategies (i.e., the $n = 0$ in particular in this study) are less susceptible to these small wind-direction variations than the WS strategy, which still has a concentrated wake on one side that is troublesome when the exact wind direction may not be known. Note that Frederik et al. (2025) also studies the $\pm 0.5D$ lateral offsets, though does so assuming that the upstream turbine is aware of the exact wind direction and is therefore actively steering the wake away from the downstream turbine for the WS case. Finally, regardless of the length of $l_y$,

the $n = -1$ case sees only a fraction of MKE recovery as does the $n = 0$ case, and this somewhat surprising result compared to literature (Frederik et al., 2020a) will be investigated further below in light of the inflow conditions.

Before continuing with the analysis, the $AP_{out}$ values from Figure 7 can be compared to the baseline-relative power improvements for the T2 turbine presented in Frederik et al. (2025) as a check-up on the capability to predict power output of a potential downstream turbine by sampling MKE. Specifically, we extract data from Figure 7(a) since the $l_y = l_z = \frac{1}{2}\sqrt{\pi}D$

control volume is positioned and dimensioned to study the flow incident on a exactly aligned hypothetical downwind turbine, and we set $x_{end} = 5D$ to correspond to the streamwise position of the downstream turbine in Frederik et al. (2025). The comparison with the results from Frederik et al. (2025) is made in Table 5 where the relative ordering of power increases for each wake-control strategy can be seen to match between the two studies. An exact match between the one- and two-turbine





simulations is not likely since the presence of two-way coupling of the downstream rotor with the flow will have an effect on
both the mean flow and the evolution of turbulent structures, and this effect will not be captured in the one-turbine simulations. It is possible to speculate, however, that part of the reason for the general positive bias of the one-turbine cases relative to the observed power increases in the two-turbine cases is due to the difference between available power (i.e., $AP_{out}$) and *useful* power for a downstream turbine. These two differ, in part, because the available flow power contained within a non-uniform flow field (i.e., sheered or veered or otherwise non-uniform) cannot be fully extracted by a wind turbine (Wagner et al., 2010). This fact may contribute to the $\approx$11% unusable power suggested by Table 5 for the WS case due to the large spatial gradient across the partially-waked downstream rotor. Other causes for discrepancies between the one- and two-turbine power predictions may include the use of a non-circular cross-section in our control volume and negligence of the streamtube expansion initiated by the downstream rotor's induction, the latter of which implies that some of the cross-sectional flow area included in the $l_y = l_z = \frac{1}{2}\sqrt{\pi}D$ control volume will not, in fact, pass through the downstream rotor disk. We do not attempt to adjust the shape or area of our original control volume, however, since an exact match cannot be achieved and since the trends in Table 5 are in satisfactory agreement with the values of Frederik et al. (2025) to suggest that our analysis of the single-turbine wakes is pertinent.

**Table 5.** Comparison of power increases relative to the baseline between the rotor-sized control volume in this article (i.e., $l_y = l_z = \frac{1}{2}\sqrt{\pi}D$) and Frederik et al. (2025). For the data from this article, $AP_{out}$ has been calculated with $x_{end} = 5D$, and the data has been adjusted to account for the varying $C_p$ of the IEA 15 MW at low wind speeds according to the $C_p$ data in Gaertner et al. (2020). Data from Frederik et al. (2025) correspond to the power uplift relative to the baseline for the downstream turbine in the fully aligned and MSLT wind condition.

| Control Strategy | % increase in $AP_{out}C_p$ | T2 uplift in Frederik et al. (2025) |
|---|---|---|
| WS | 78.78 % | 67.93 % |
| WM, $n = 0$ | 33.14 % | 34.66% |
| WM, $n = -1$ | 24.71 % | 23.09% |
| WM, $n = \pm 1$ ($cl = 90^o$) | 16.34 % | 16.03% |
| WM, $n = \pm 1$ ($cl = 0^o$) | 14.62 % | n/a |

## 3.2 Accounting of MKE budget

### 3.2.1 Verification of technique

Before leveraging the control-volume analysis to understand the terms contributing to recovery of MKE, we first verify the methodology. The main goal of this verification is to confirm that the LHS and RHS of the transport equation for axial MKE (i.e., Eq. (5) and Eq. (6)) are sufficiently equal, which will offer confidence that the formulation of the MKE budget is proper as well as that the terms of the equations are being sampled appropriately and calculated correctly.



The results of the verification are shown in Table 6. The residual is always less than 1% of $AP_{out}$, which is better than the
2% level of accuracy reported by Calaf et al. (2010) in their MKE budget of non-actuated wakes. Calaf et al. (2010) attributed
the remaining error to incomplete statistical convergence, and we similarly conclude from our small residuals that the dominant
mechanisms of MKE transport are being adequately captured. It is noted that, from among the terms in Eq. (6) that contribute
to $AP_{in}$, the exclusion of the $\mathcal{W}_{bf}$ terms discussed previously affects the residuals in Table (6) by less than $\pm0.5\%$, and thus
the terms related to Coriolis forcing and compensation for blockage are ignored for the remainder of the analysis. Note also
that for the larger control volume to be considered below (i.e., $l_y = (\frac{1}{2}\sqrt{\pi} + 1)D$), only $yz$ planes and not $xy$ nor $xz$ planes
were available, but the residuals remained below 1% even when using the streamwise-interpolated $yz$ planes to calculate the
surface fluxes on the four side faces.

**Table 6.** Contributions to the MKE budget for $x_{end} = 5D$ and $l_y = l_z = \frac{1}{2}\sqrt{\pi}D$. Data are nondimensionalized on $l_y l_z U_{hh}^3$.

| Control Strategy | $AP_{out}$ (Eq. 5) | $AP_{in}$ (Eq. 6) | Residual (% of $AP_{out}$) |
|---|---|---|---|
| Baseline | 0.1750 | 0.1741 | -0.0009 (-0.50%) |
| WS | 0.2924 | 0.2928 | 0.0004 (0.13%) |
| WM, $n = 0$ (pulse) | 0.2249 | 0.2238 | -0.0011 (-0.50%) |
| WM, $n = -1$ (ccw helix) | 0.2121 | 0.2109 | -0.0012 (-0.58%) |
| WM, $n = \pm1$, $cl = 90°$ (side-side) | 0.1993 | 0.1983 | -0.0010 (-0.52%) |
| WM, $n = \pm1$, $cl = 0°$ (up-down) | 0.1967 | 0.1957 | -0.0010 (-0.52%) |

### 3.2.2 Case 1: Aligned wind direction

Having determined that $AP_{in}$ is an acceptable estimate of $AP_{out}$, this section traces the source of the improved MKE recovery
by the wake-control strategies to the specific terms that compose $AP_{in}$, and this is done for the smaller control volume (i.e.,
$l_y = l_z = \frac{1}{2}\sqrt{\pi}D$) that might be considered the inflow region for a downstream turbine when the wind direction is completely
aligned with the turbine column. Table 7 provides the contributions of each term in Eq. 6 (excluding the $\mathcal{W}_{bf}$ terms, as
discussed) to $AP_{in}$ for the control volume ending at $x_{end} = 5D$.

First, the WS case is considered. The source of the increase in $AP_{in}$ for WS lies with the $\phi_{mean}$ terms and especially
$\phi_{mean,lf}$, which increases by 44.0% relative to the baseline as it benefits from ambient wind replacing the flow being steered
out the other side of the control volume. The effect on the opposing $\phi_{mean,rg}$ is opposite as the right face overlaps with more
wake outflow because of the steering motion, and $\phi_{mean,rg}$ decreases. An interesting observation is the 33.6% gain by $\phi_{mean,tp}$
that will be the subject of discussion in Sect. 3.3.1. It is also evident from the large value of $\phi_{mean,fr}$ that the yawed rotor
extracts less energy from the flow, as expected, and this higher initial MKE in the case of WS is at least partially offset by
lower rotor power as confirmed in Frederik et al. (2025).





**Table 7.** Contributions to $AP_{in}$ relative to the baseline case for $x_{end} = 5D$ and $l_y = l_z = \frac{1}{2}\sqrt{\pi}D$. Values shown are the difference from the baseline per term as a percentage of the total $AP_{in}$ from the baseline. The bold face values are the largest positive contributions to the last column of each row.

| Control Strategy | $\phi_{mean}$ | | | | | $\phi_{turb}$ | | | | | $\mathcal{P}$ | $\mathcal{W}_p$ | $AP_{in}$ |
|---|---|---|---|---|---|---|---|---|---|---|---|---|---|
| | front (fr) | left (lf) | right (rg) | top (tp) | bottom (bt) | front (fr) | left (lf) | right (rg) | top (tp) | bottom (bt) | | | |
| WS | 25.6% | **44.0%** | -25.1% | 33.6% | -6.8% | 0.3% | 0.1% | -2.3% | -2.6% | 1.0% | 1.5% | 0.2% | 68.2% |
| WM, $n = 0$ (pulse) | -2.8% | 2.8% | 3.8% | -10.5% | -0.1% | 2.3% | 15.9% | 2.0% | **20.6%** | 1.7% | -3.8% | -3.4% | 28.5% |
| WM, $n = -1$ (ccw helix) | 2.3% | 1.0% | -3.8% | 5.9% | 1.2% | 2.3% | **7.9%** | 3.0% | 3.7% | 0.2% | -1.0% | -1.3% | 21.1% |
| WM, $n = \pm1$, $cl = 90^o$ (side-side) | 1.4% | -3.0% | -4.7% | 4.8% | 1.4% | 1.1% | **6.7%** | 4.1% | 3.8% | 0.9% | -1.7% | -0.8% | 13.9% |
| WM, $n = \pm1$, $cl = 0^o$ (up-down) | 1.1% | 3.6% | -1.2% | 0.6% | 0.1% | 1.0% | 0.7% | **5.0%** | 2.0% | 0.2% | -0.3% | -0.2% | 12.4% |

Unlike the WS case, the WM cases see smaller variations in $\phi_{mean}$ from the baseline and larger gains from $\phi_{turb}$. Notably, for the $n = 0$ case, the $\phi_{turb,lf}$ and $\phi_{turb,tp}$ account for almost all the increases over the baseline $\phi_{turb}$ terms with increases of 15.9% and 20.6%, respectively. For the other WM cases, this trend of relatively large gains from $\phi_{turb,lf}$ and $\phi_{turb,tp}$ holds, but the magnitudes of increase are notably smaller. The $n = \pm1(cl = 90^o)$ case, which features side-to-side movement, intuitively

sees the improvement in $\phi_{turb,rg}$ rise above that of $\phi_{turb,tp}$. Interestingly, the up-and-down motion of the $n = \pm1(cl = 0^o)$ case does *not* have its largest $\phi_{turb}$ increases on the top or bottom faces but rather on the right face. This may be related to a finding in Cheung et al. (2024) and Yalla et al. (2025a) indicating that the up-and-down forcing of the rotor is converted into an axisymmetric mode rather than producing a strong flapping motion. It is noted also that the $\mathcal{P}$ term always decreases for the WM cases to the detriment of the MKE recovery, and this is intuitive since the intentional increase in turbulence and Reynolds

shear stress for the WM strategies will result in additional production.

Figure 8 shows the same terms as Table 7 but highlights their evolution in the streamwise direction by plotting their gradients with respect to $x/D$. First, the baseline case is considered in panel (a), where most of the near-wake loss in MKE stems from the $\mathcal{W}_p$ and $\phi_{mean}$ terms as the wake streamtube completes its expansion. Starting from $x/D \approx 1.5D$, the MKE begins recovering, and the mean-flow convection takes on a dominant role in this process, a fact that contrasts with the results of Boudreau and

Dumas (2017) where turbulent transport was dominant. This difference is a result of the high veer in the present study's inflow that causes $\phi_{mean,lf}$ to be the dominant contributor to recovery everywhere outside the near wake. Besides $\phi_{mean,lf}$, it is the top surface that is generally responsible for the largest MKE gains in terms of both mean-flow and turbulent transport.

Similar observations as well as several new ones apply to the wake-control cases. The strong increases in $\phi_{mean,lf}$ and $\phi_{mean,tp}$ reported above for the WS case are observed in Figure 8(b) to be consistent for all locations downstream of $x/D$=1.

Similarly, the previously mentioned dominance of the $\phi_{turb,lf}$ and $\phi_{turb,tp}$ terms in the recovery of the $n = 0$ and $n = -1$





cases is consistent across most of the streamwise locations except at $x/D < 2$ where the contributions to $\phi_{turb}$ from all four faces are significant. It can also be observed that the wake-control strategies shift the $x/D$ location of maximum recovery rate forward to around 2-4$D$ versus the location of the baseline's maximum at 4-5$D$.



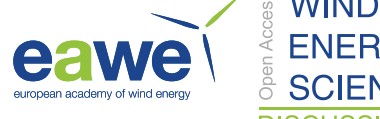



**Figure 8.** Gradient of the budget of MKE terms of $AP_{in}$ (i.e., Eq. 6) with respect to nondimensional streamwise location. Each term is nondimensionalized on $U_{hh}^3 l_z$, and $l_y = l_z = \frac{1}{2}\sqrt{\pi}D$.





### 3.2.3 Case 2: Varying wind direction

This section examines the MKE characteristics of the wake-control strategies for the case of a control volume with a wider cross-sectional area, which has relevance when the wind is not exactly aligned with a turbine column such as due to imperfect knowledge of the wind direction as discussed previously related to OUU. Table 8 and Figure 9 corresponds to Table 7 and Figure 8, respectively, but for the case of the wider control-volume dimensions of $l_y = (\frac{1}{2}\sqrt{\pi}+1)D$. One obvious difference of Table 8 compared to Table 7 is the overall smaller magnitudes of values, which result from the wake occupying a smaller

fraction of the control volume. In terms of $\phi_{mean}$, the left and right faces still play an important role in the MKE changes of the wake-control strategies for the larger control volume, however, these faces take on much less importance for $\phi_{turb}$ because of the farther separation of these faces from the wake itself. Therefore, there is increased opportunity for turbulent MKE recovery from aloft through the top face as discussed in Sect. 1 related to the literature on larger-array wind farms. In this regard, the WM case of $n = 0$ leads with 8.3% improvement in $\phi_{turb,tp}$ over the baseline as compared to -0.9% for the WS case. This 8.3%

improvement might not be enough alone to produce the high MKE recovery of the $n = 0$ case in Fig.7(b), but the $\phi_{mean,tp}$ term also contributes a 5.1% improvement over the baseline for this case. Thus, while the $\phi_{mean,tp}$ has not conventionally been expected to contribute to the MKE recovery in the limit of a wide control volume (or wind farm), there is opportunity for local pockets of vertical circulation of mean flow that produce appreciable recovery.

**Table 8.** Contributions to $AP_{in}$ relative to the baseline case for $x_{end} = 5D$, $l_y = (\frac{1}{2}\sqrt{\pi}+1)D$, and $l_z = \frac{1}{2}\sqrt{\pi}D$. Values shown are the difference from the baseline per term as a percentage of the total $AP_{in}$ from the baseline. The bold face values are the largest positive contributions to the last column of each row.

| Control Strategy | $\phi_{mean}$ | | | | | $\phi_{turb}$ | | | | | $\mathcal{P}$ | $\mathcal{W}_p$ | $AP_{in}$ |
|---|---|---|---|---|---|---|---|---|---|---|---|---|---|
| | front | left | right | top | bottom | front | left | right | top | bottom | | | |
| WS | 9.8% | **10.9%** | -8.9% | 9.3% | -0.4% | 0.1% | -0.2% | 0.3% | -0.9% | -0.2% | 2.0% | 3.1% | 23.9% |
| WM, $n = 0$ (pulse) | 0.8% | 5.8% | 1.9% | 5.1% | -0.2% | 1.1% | 0.2% | 0.0% | **8.3%** | 0.9% | -2.0% | -1.6% | 20.1% |
| WM, $n = -1$ (ccw helix) | 2.1% | **3.2%** | -0.4% | 1.6% | 0.3% | 0.9% | 0.2% | 0.1% | 2.4% | 0.3% | -0.7% | -0.1% | 9.7% |
| WM, $n = \pm 1$, $cl = 90^o$ (side-side) | 1.6% | **2.6%** | -1.3% | 1.8% | -0.2% | 0.5% | 0.2% | 0.3% | 1.9% | 0.5% | -1.2% | 0.3% | 6.9% |
| WM, $n = \pm 1$, $cl = 0^o$ (up-down) | 1.6% | **2.4%** | 0.0% | 1.3% | 0.6% | 0.4% | -0.1% | 0.7% | 1.0% | 0.0% | -0.3% | 0.2% | 7.8% |

In Fig. 10, we therefore track the streamwise development of the $\phi_{mean,tp}$ and $\phi_{turb,tp}$ terms considering the wider control

volume. As in Table 8, the improved performance along the top face of the WS case over the $n = 0$ case in terms of $\phi_{mean,tp}$ is more than offset by the improved performance of the latter over the former in terms of $\phi_{turb,tp}$ for all $x/D$ locations outside the near wake. This result directly motivates the analysis in the next section.


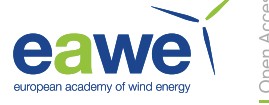
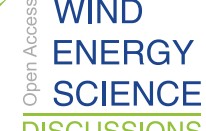

**Figure 9.** Gradient of the budget of MKE terms of $AP_{in}$ (i.e., Eq. 6) with respect to nondimensional streamwise location. Each term is nondimensionalized on $U_{hh}^3 l_z$, $l_y = (\frac{1}{2}\sqrt{\pi} + 1)D$, and $l_z = \frac{1}{2}\sqrt{\pi}D$.

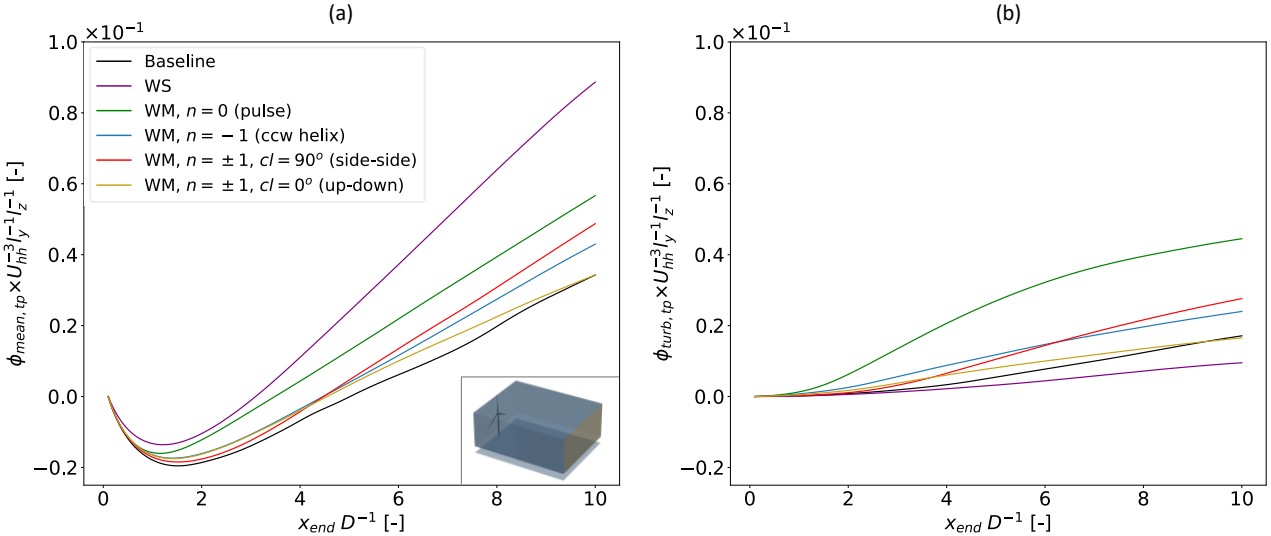

**Figure 10.** Cumulative change in the normalized (a) $\phi_{mean,tp}$ and (b) $\phi_{turb,tp}$ terms of the MKE budget for the wider control volume with $l_y = (\frac{1}{2}\sqrt{\pi} + 1)D$ and $l_z = \frac{1}{2}\sqrt{\pi}D$.

### 3.3 Detailed analysis of top surface

Considering the real-world prevalence of larger-array wind farms and the need for re-energizing in such farms through the top surface of the control volume as discussed in Sect. 1, we now focus the analysis on mean convection and turbulent transport through this top surface. The goal is to pinpoint the origin of the changes in MKE recovery for the wake-control strategies versus the baseline in Fig. 10. The following two subsections therefore investigate the spatial distribution of the quantities contributing to $\phi_{mean,tp}$ and $\phi_{turb,tp}$.

#### 3.3.1 Mean convection through top surface

Fig. 11 shows contours of $-\overline{u}^2\overline{w}$, the quantity that is integrated along the top surface of the control volume to produce $\phi_{mean,tp}$. Across all cases, the dominant sign of $-\overline{u}^2\overline{w}$ outside of the rotor and near-wake region is positive (i.e., flux into the control volume) as mean flow from aloft is drawn down to replenish the rotor layer. The same quantity is plotted in Fig. 12 along cross-stream planes at $x=3D$ for select cases.

The WS case exhibits both increases in $\phi_{mean,tp}$ compared to the baseline at more positive values of $y$ as well as slight decreases in the same at more negative values of $y$. Insight on the cause for these changes can be gathered from Fig. 12(b), which shows firstly that the wake deflects towards negative $y$ and positive $z$ compared to the baseline, as expected according to Bastankhah and Porté-Agel (2016). The lateral velocity components also indicate a possible exaggeration of the counterclockwise vorticity near the wake center compared to the baseline, suggesting that WS is augmenting the natural swirl of the wake,



possibly due to merging of the upper vortex of the CVP with the main wake swirl of same handedness. The circulation of this

augmented vortex produces a region of positive $\overline{u}^2\overline{w}$ compared to the baseline at $y/D \approx$ -1 and of negative $\overline{u}^2\overline{w}$ centered at $y/D \approx 0.25$ in Fig. 12(b). As evidenced by comparing Tables 7-8 and in agreement with the understanding of yaw-added wake recovery described in Sect. 1, the larger magnitude of the $\overline{u}^2\overline{w}$ gain over the top surface compared to the loss results in a net increase to $\phi_{mean,tp}$ from WS.

For the WM cases, the larger $\phi_{mean,tp}$ observed in Fig. 10(b) is related to enhancement of the region of positive $\overline{u}^2\overline{w}$ at

$y/D \approx 0.5$ compared to the baseline. The $n = 0$ case shows a further modification from the baseline that is evident in Fig. 12(c): a second lobe of negative $\overline{u}^2\overline{w}$ at $y/D \approx -0.75$. The presence of this second lobe may be related to another interesting feature of the $n = 0$ case, which is its apparent resistance to skewing from veer until at least $x/D \approx 5$. Examining Fig. 12(c), the primary swirl structure of the wake visible in panels (a) and (b) is already nonexistent by $x/D = 3$, suggesting that the $n = 0$ case is breaking down the original wake structure relatively quickly and allowing for the $\overline{u}^2\overline{w}$ on the top surface of the control

volume at $y/D \approx -0.75$, which is normally governed by the swirl of the wake, to be directed into rather than out of the control volume. Mass continuity within the control volume is conserved in part by a small mean outflow from the top surface at $y \approx 0$. However, the inflow from aloft has higher momentum than this outflow from the wake. This effective mean-recovery mechanism also appears in simulations of larger arrays of turbines for the $n = 0$ case with similar inflow conditions in Yalla et al. (2025b), and its relation to the turbulent structures in the wake should be considered. As seen by comparing the $n = 0$

data from Fig. 10(a) and (b), this improved recovery of $\phi_{mean,tp}$ accounts for a sizable fraction of the overall improvement in MKE recovery along the top surface compared to the baseline for the $n = 0$ case, and it could also suggest that the WM case of $n = 0$, at least, might benefit downstream turbines that are offset in one lateral direction more than the other though this hypothesis should be investigated with more inflow conditions.

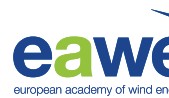


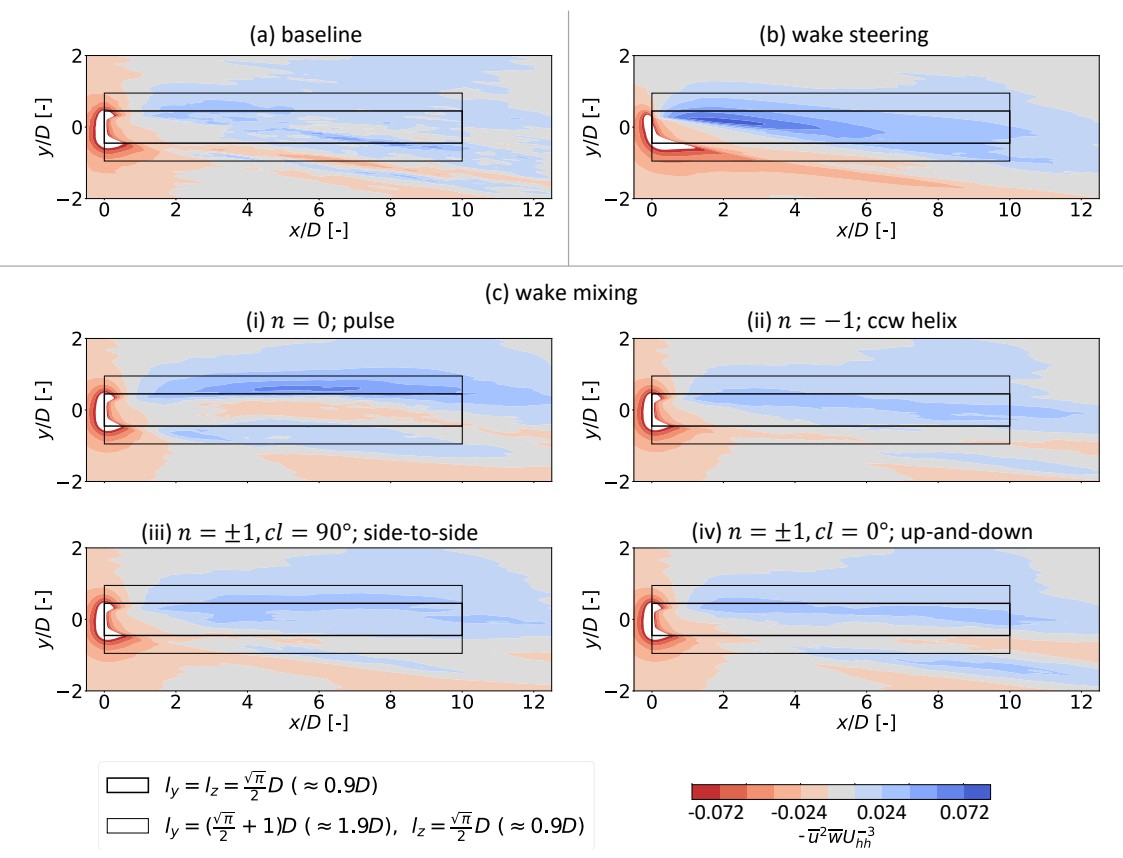

**Figure 11.** Contours of normalized $-\overline{u}^2\overline{w}$ along the $xy$ plane at $z = \frac{1}{4}\sqrt{\pi}D$, or $\approx 0.44D$ (i.e., the top surface of the control volume). Positive (blue) values indicate MKE flux into the control volume from aloft. The flow is viewed from above looking down.



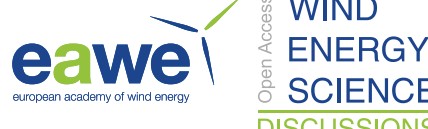

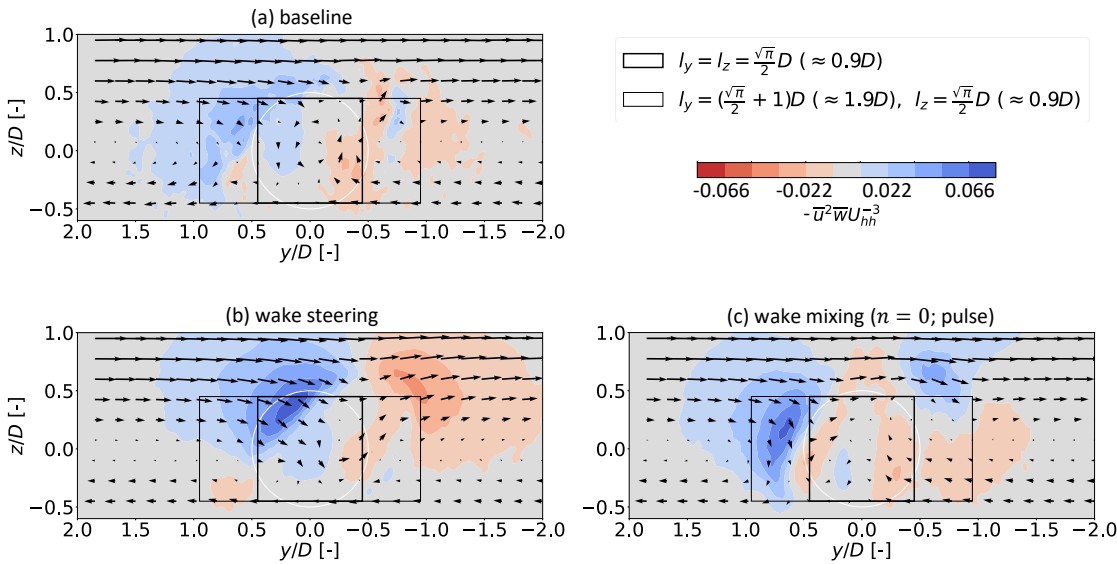

**Figure 12.** Contours of normalized $-\overline{u}^2\overline{w}$ along the $yz$ plane at $x = 3D$. Positive (blue) values along the top face indicate flux into the control volume. Vectors are composed of $\overline{v}$ and $\overline{w}$ and have the same scaling based on magnitude in all panels. The flow is viewed from upstream looking downstream.





### 3.3.2 Turbulent entrainment through top surface

Fig. 13 corresponds to Fig. 11, except it shows the distribution of turbulent entrainment (i.e., $\overline{u}\overline{u'w'}$) rather than mean convection (i.e., $\overline{u}^2\overline{w}$) along the top surface of the control volume, and Fig. 13 thus informs the behavior of the $\phi_{turb,tp}$ in Fig. 10. Across all cases, the sign of -$\overline{u}\overline{u'w'}$ nearly everywhere in the sampling domain is positive (i.e., flux into the control volume) as the mean shear of the wake produces turbulent entrainment of higher-velocity ambient flow aloft into the lower-velocity wake flow below. Fig. 13(b) confirms the observation from Fig. 10(b) that the wake of a turbine using WS does not benefit

from improved $\overline{u'w'}$ at top tip; in fact, it decreases it compared to the baseline. The WM cases, on the other hand, show improvements relative to the baseline, which is expected given the unsteady nature of the WM strategies.

In light of this unsteady, periodic nature of the WM strategies, a phase-averaged analysis of the flow fields is useful to pinpoint the source of the increases in -$\overline{u}\overline{u'w'}$ observed in Fig. 13. Our phase-averaged analysis follows from Lignarolo et al. (2015) and defines the phase-averaged (alternatively known as phase-locked) contribution of the turbulent fluxes to MKE

recovery by $\overline{u}(\tilde{u}_\varphi \tilde{v}_\varphi)$ and $\overline{u}(\tilde{u}_\varphi \tilde{w}_\varphi)$ where

$$\tilde{u}_\varphi = \langle u \rangle_\varphi - \overline{u}; \quad \tilde{v}_\varphi = \langle v \rangle_\varphi - \overline{v}; \quad \tilde{w}_\varphi = \langle w \rangle_\varphi - \overline{w} \tag{8}$$

and where the $\langle \cdot \rangle_\varphi$ operator refers to quantities averaged over a particular phase, $\varphi$, of the Strouhal period for the whole simulation length. We can define phase-averaged versions, $\phi_{turb\text{-}p_\varphi}$, of the $\phi_{turb}$ terms in Eq. (6) as in Eq. (9)

$$\phi_{turb\text{-}p_\varphi} = \pm \iint\limits_{tp,bt} \overline{u}(\tilde{u}_\varphi \tilde{w}_\varphi) \, dS \pm \iint\limits_{lf,rg} \overline{u}(\tilde{u}_\varphi \tilde{v}_\varphi) \, dS \tag{9}$$

For the analysis below, we apply Eq. (9) to the control volume with cross-sectional side lengths of $l_y = l_z = \frac{1}{2}\sqrt{\pi}D$. The $\phi_{turb\text{-}p_\varphi}$ terms have been calculated for 16, equally-spaced $\varphi$ angles: [0°:22.5°:337.5°]. Each angle is averaged over the 14 cycles of Strouhal period of 88.59 s (i.e., *not* the rotor period of <8 s). It should be noted that the limited number of Strouhal cycles introduces non-negligible uncertainty to the phase-averaged comparisons.

Figure 14(a) plots the streamwise derivatives of the $\phi_{turb\text{-}p_\varphi}$ terms versus $\varphi$ at $x/D = 3$ for the $n = 0$ case. The periodic

nature of the MKE recovery is apparent from a peak near $\varphi = 180°$. Figure 14(b), plotting the corresponding cross-sections with $\tilde{u}$, $\tilde{v}$, and $\tilde{w}$, shows the cause for this peak is a mushroom-like ejection of fluid with lower phase-averaged mean-flow energy from the top, left, and right sides of the volume, as well as a sweeping inflow near the bottom of the left and right sides. The impact of the mushroom-like structure can be observed throughout the progression of phases shown in Figure 14(b), and its effect on the plane at $x/D = 3$ is related to the relative position of the convecting vortex ring depicted in Figure 14(c). In fact,

$\varphi = 180^o$ is the phase when this ring passes the plane, and its effect on MKE recovery for the top, left, and right sides switches from sweeps for $\varphi < 180^o$ to ejections for $\varphi > 180^o$. A trough in the MKE recovery occurs at $\varphi = 360^o$, and this corresponds to the $x/D = 3$ plane being midway between consecutive ring vortices as shown in Figure 14(c)(i). At this phase, the induced



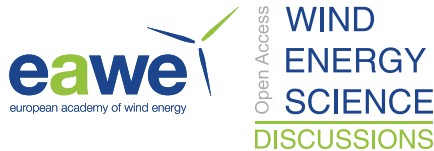

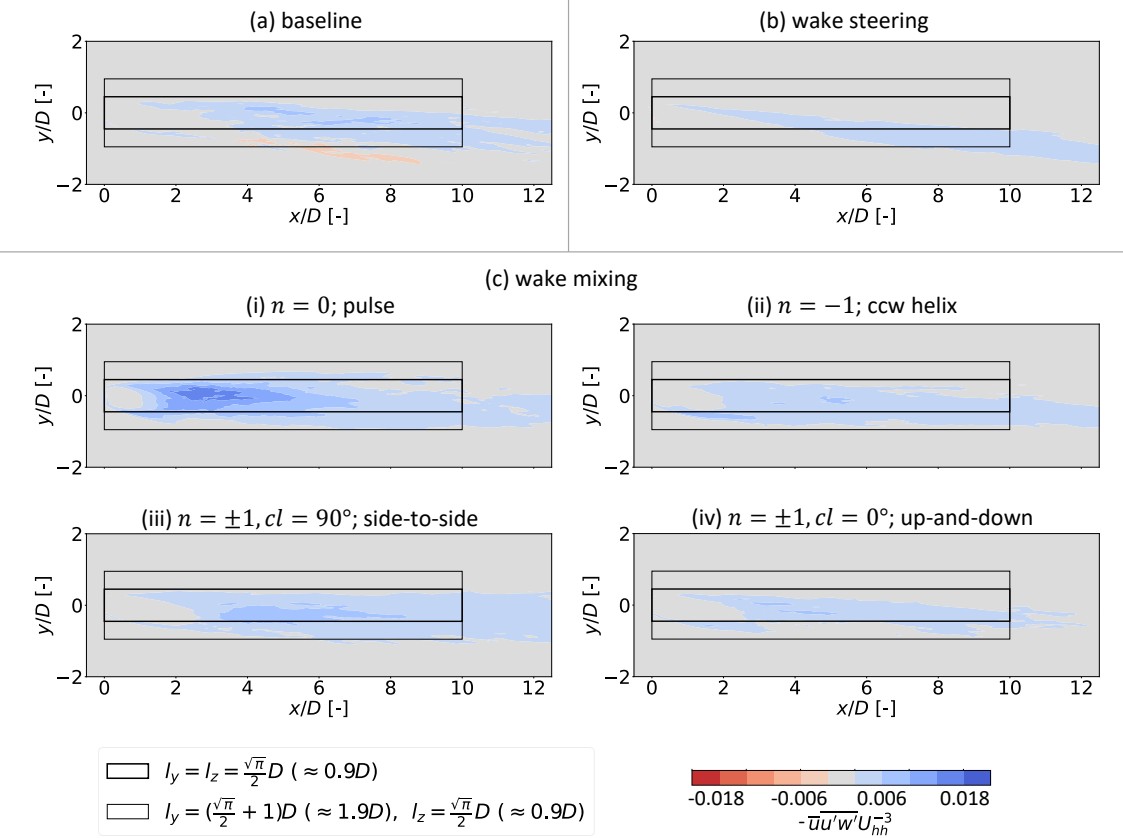

**Figure 13.** Contours of normalized $-\overline{u}\overline{u'w'}$ along the $xy$ plane at $z = \frac{1}{4}\sqrt{\pi}D$, or $\approx 0.44D$ (i.e., the top surface of the control volume). Positive (blue) values indicate MKE flux into the control volume from aloft. The flow is viewed from above looking down.

lateral velocities at $x/D = 3$ from the ring vortices are counteracting. However, the total $\partial\phi_{turb\text{-}p_\varphi}/\partial x$ remains slightly above zero due to momentum entering the control volume from the top face.

Figure 15 shows the same plots as Figure 14 but for the $n = -1$ case. The most prominent $\phi_{turb\text{-}p_\varphi}$ term in Figure 15(a) alternates between the four sides of the control volume according to the helically-winding thrust pattern produced by the blades. This alternation is also apparent in panel (b) where the region of lower-momentum $\tilde{u}$ rotates counterclockwise from 8 o'clock to 5 to 2 to 11 in subpanels (i-iv), respectively. Insight is provided from the visualizations in Figure 15(c) where it can be seen that the rotating region of lower $\tilde{u}$ coincides with the passage of the helical vortex structure (identifiable by an amalgamation of

slightly darker isosurfaces) through the plane. This follows from the understanding that the helical vortex structure derives from the high shear condition at the rotor tip created by the phase of the Strouhal cycle with maximum blade loading (i.e., minimum $\tilde{u}$ in the wake). The $\tilde{v}$ and $\tilde{w}$ are the quantities that dictate the flux of $\phi_{turb\text{-}p_\varphi}$ across the faces of the control volume, and these indicate that MKE recovery is generally occurring at both the location of minimum $\tilde{u}$ described above through ejections, as




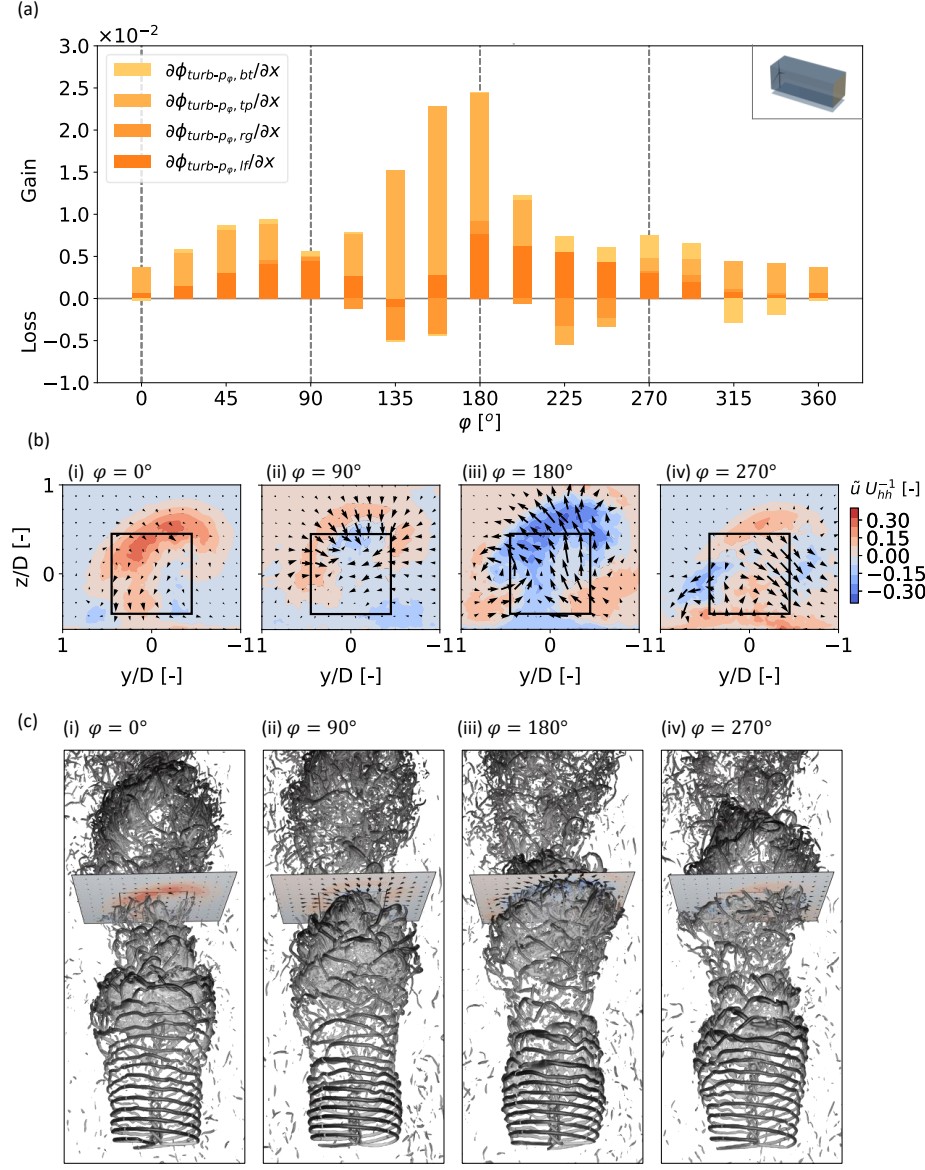

**Figure 14.** Phase-averaged analysis of the $n = 0$ case (pulse) at $x/D = 3$ including (a) the normalized streamwise derivatives of $\phi$ terms for phase-averaged turbulent transport using the control volume with $l_y = l_z = \frac{1}{2}\sqrt{\pi}D$, (b) phase-averaged cross sections of the flow field for four of the phase angles shown in (a), and (c) visualization of iso-contours of vorticity colored by streamwise velocity and viewed from above the wake. In (b), the vectors are $\tilde{v}$ and $\tilde{w}$ and have the same scaling according to magnitude in all sub panels, and the flow is viewed from downstream looking upstream. The same flow cross section at $x/D = 3$ is seen in both (b) and (c).



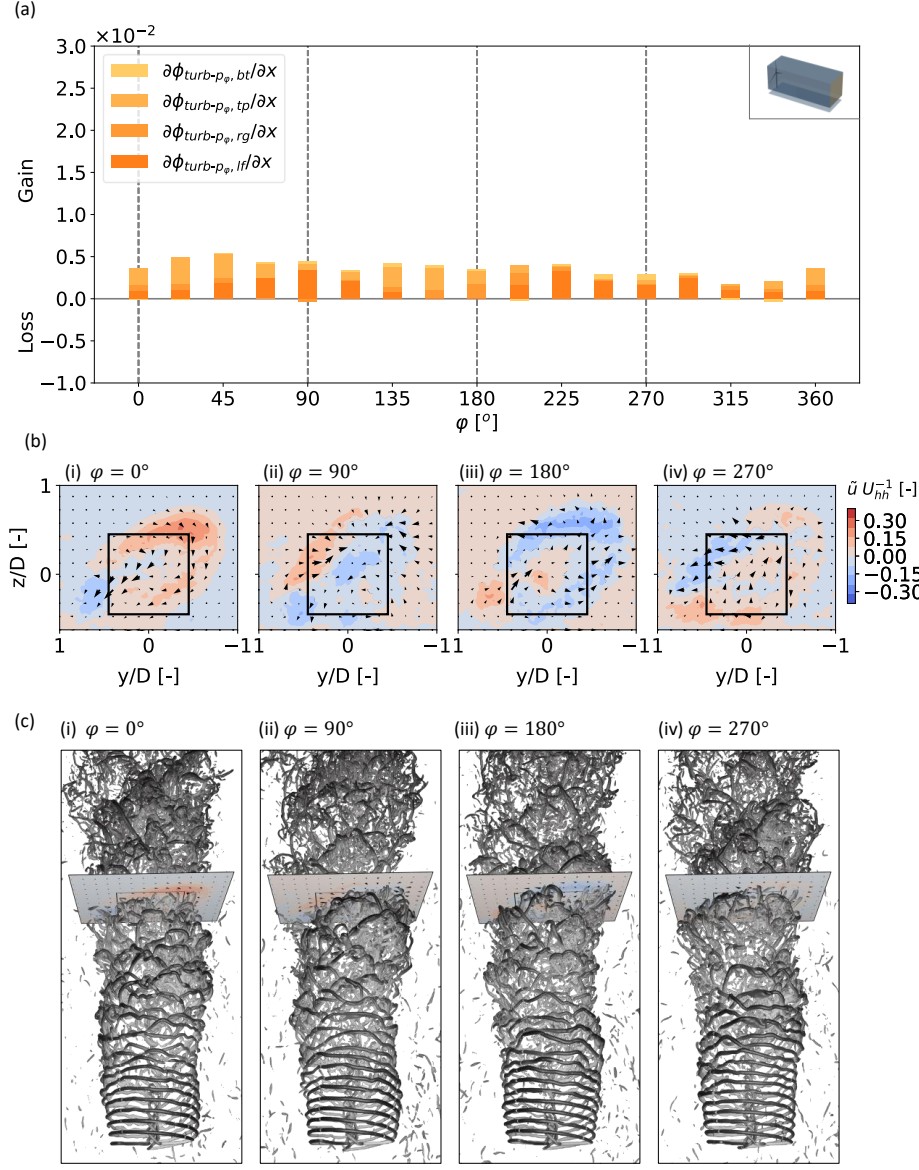

**Figure 15.** Phase-averaged analysis of the $n = -1$ case (ccw helix) at $x/D = 3$ including (a) the normalized streamwise derivatives of $\phi$ terms for phase-averaged turbulent transport using the control volume with $l_y = l_z = \frac{1}{2}\sqrt{\pi}D$, (b) phase-averaged cross sections of the flow field for four of the phase angles shown in (a), and (c) visualization of iso-contours of vorticity colored by streamwise velocity and viewed from above the wake. In (b), the vectors are $\tilde{v}$ and $\tilde{w}$ and have the same scaling according to magnitude in all sub panels, and the flow is viewed from downstream looking upstream. The same flow cross section at $x/D = 3$ is seen in both (b) and (c).





well as at an opposing azimuth position through sweeps. Despite this twice per Strouhal cycle influx of MKE on each face
of the control volume, it is clear from the lower magnitudes of $\partial\phi_{turb\text{-}p_\varphi}/\partial x$ that the WM mechanism of the $n = -1$ case is
not as effective as that utilized by the $n = 0$ case. Notably, the $n = -1$ case fails to induce the large magnitude of $\tilde{v}$ and $\tilde{w}$
perturbations seen in the $n = 0$ case.

Both Figs. 14 and 15 portray turbulent transport processes precipitated by the passing of periodic, coherent flow structures
at $x/D = 3$. While our phase-averaged control-volume analysis cannot distinguish between the periodic spreading of the wake
deficit due to deflection and the periodic entrainment from passing vortices, it is noted that the former process should be most
prominent in the near wake. Thus, the significant contributions to MKE recovery for the WM cases that continue to $x/D = 3$
and beyond as shown in Fig. 8 may be a result of vortex-induced entrainment. A hypothesis is thus that the $\phi_{turb}$ benefit of the
$n = 0$ and $n = -1$ (and other WM) strategies in our simulations is as much or more a result of ongoing sweeps and ejections
from the passing of periodic flow structures as it is from the spreading of the near wake over a wider cross-sectional area. This
motivates the characterization of these coherent structures as performed in Cheung et al. (2024) and Yalla et al. (2025a), as
well as the investigation of what wake conditions cause such structures to be most amplified.

### 3.4 Effect of veer and turbulence

The MSLT condition discussed throughout this article represents a wind condition that is believed to be favorable for wake-
control technology because of the longevity of wakes in such low turbulence conditions. According to the simple binning
described in Sect. 2.3, the wind condition with the next highest turbulence level (and with significantly less veer), MSMT, also
occurs with relative frequency, and wake control may still be relevant for this case. Indeed, Frederik et al. (2025) examined
the MSMT condition and found some wake-control strategies produce $\approx 5\%$ uplift in power for the aligned, two-turbine array,
which is not far from the $\approx 7\%$ maximums for the MSLT condition. This subsection is therefore devoted to comparing and
contrasting our results related to control-volume analyses between the MSLT and MSMT conditions.

It must be mentioned that the residual after subtraction of Eq. (5) from Eq. (6) was slightly larger for the MSMT case; it
was on the order of 1-3% compared to <1% reported previously for the MSLT case. Two reasons for this difference may be the
higher ambient turbulence in the MSMT case that requires longer sampling time to reach converged statistics and the use again
of the streamwise-interpolated $yz$ planes rather than dedicated $xy$ and $xz$ planes to calculate the surface fluxes on the four side
faces as explained previously for the wider control volume in the MSLT analysis. Further, $AP_{out}C_p$ at $x_{end} = 5D$ from the
MSMT condition shows higher variation with the T2 uplift reported in Frederik et al. (2025) than for the MSLT condition; the
$AP_{out}C_p$ at $x_{end} = 5D$ for the WS and WM $n = -1$ case to be examined below show absolute differences of 10-20 percentage
points from the corresponding values in Frederik et al. (2025). As a result of these limitations, this section shows the qualitative
trends between the MSLT and MSMT cases while not showing the depth of quantitative values presented above for the MSLT
case.

To this end, the results in Frederik et al. (2025) show two main differences between the power for the downstream turbine
(T2) at the MSMT condition compared to that of the MSLT one: (1) the WS case exhibits $\approx 7\%$ lower power for T2 compared





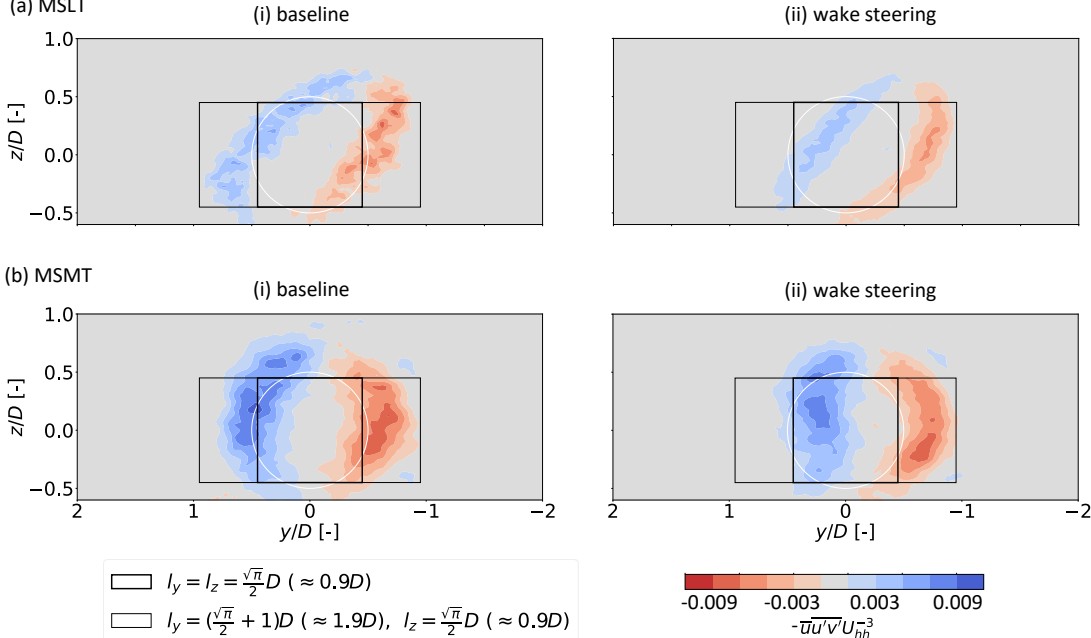

**Figure 16.** Contours of normalized $-\overline{u}\overline{u'v'}$ along the $yz$ plane at $x = 3D$. Positive (blue) values along the left face indicate flux into the control volume as do negative (red) values along the right face. The flow is viewed from upstream looking downstream.

to the baseline and (2) the $n = -1$ case shows $\approx 6\%$ higher power compared to the baseline, performing on par with the $n = 0$ case.

The reduced effectiveness of WS at the MSMT condition is related to the increased relative importance of turbulent transport
in the MKE budget for the MSMT condition. This is illustrated by the comparison of Fig. 16(a) and (b), which show contours of $\overline{u}\overline{u'v'}$ (i.e., the quantity which integrates to form $\phi_{turb,lf}$ and $\phi_{turb,rg}$). For the MSLT condition in (a), $\overline{u}\overline{u'v'}$ has relatively small magnitude, and, because of the skew angle of the wakes, the action of steering the wake does little to change the net transport through the left and right faces. However, in the MSMT condition $\overline{u}\overline{u'v'}$ (and $\overline{u}\overline{u'w'}$) take on larger magnitudes and importance in the MKE balance due to the high ambient turbulence intensity. Furthermore, the absence of significant skewing
of the wake sees the region of turbulent transport at the left and right edges of the wake have significant overlap with the left and right faces of the control volume, respectively, for the baseline case. On the other hand, the action of steering the wake moves the region of peak negative $\overline{u}\overline{u'v'}$ into the center of the wake where it has little impact on the surface flux at the left face of the control volume. Similarly, the peak of positive $\overline{u}\overline{u'v'}$ is steered off to negative $y$ values that are not as relevant for the surface flux at the right face of the control volume. Thus, the correlated change in turbulence intensity and inflow veer produces
a negative effect for WS.

The increased effectiveness of the $n = -1$ case at the MSMT condition may be related, in part, to an improvement in $\phi_{turb}$ compared to that of the MSLT condition. Given the usefulness of phase-averaged data to explain differences in turbulent





entrainment for the WM cases, Fig. 17 compares the streamwise development of $\partial\phi_{turb\text{-}p}/\partial x$ for the MSLT and MSMT conditions. Note that $\phi_{turb\text{-}p}$ is the mean over all phases and thus represents the total phase-averaged contribution to $\phi_{turb}$

(though not the total $\phi_{turb}$, which also has contributions from non-phase-locked fluctuations). It is apparent that the $n = -1$ case shows marked increases in $\partial\phi_{turb\text{-}p}/\partial x$ up to $x/D = 5$ for the MSMT case, though this increase may be exaggerated compared to the two-turbine results of the companion paper (Frederik et al., 2025). This behavior can be examined more closely in Fig. 18, which shows phase-averaged cross sections at $x/D = 3$ from the MSLT and MSMT conditions (Fig. 18(a) is identical to Fig. 15(b) for convenient side-by-side comparison). In the (unskewed) MSMT case in panel (b), the $n = -1$

instability mechanism is shown to be effective; the troughs of $\tilde{u}$ are ejected out of the control volume while the peaks of $\tilde{u}$ are swept into it. It is proposed that the non-skewed wake shape of the MSMT condition produces a stronger coherent structure as this mechanism relies on spatial proximity to the helical windings of adjacent phases, and this proximity is reduced as the wake is skewed by inflow veer. Similar logic might *not* hold for the $n = 0$ case, which apparently has the effect of reducing wake skew at least over the $x/D$ ranges most relevant for wake recovery as shown in Fig. 6. This could explain, in part, why

the performance of the $n = 0$ case as reported in Frederik et al. (2025) varied much less than that of the $n = -1$ case between the MSLT and MSMT conditions.

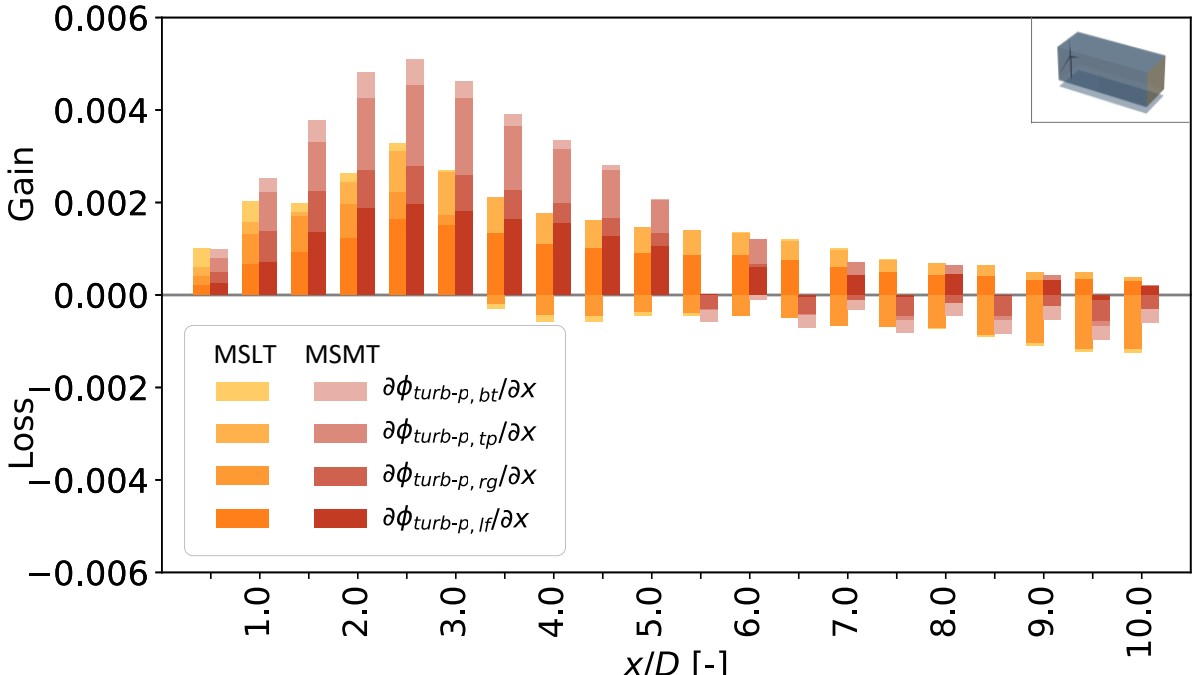

**Figure 17.** Streamwise development of $\partial\phi_{turb\text{-}p}/\partial x$ terms for the $n = -1$ case (ccw helix) from both the MSLT and MSMT conditions using the control volume with $l_y = l_z = \frac{1}{2}\sqrt{\pi}D$.

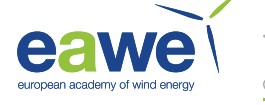


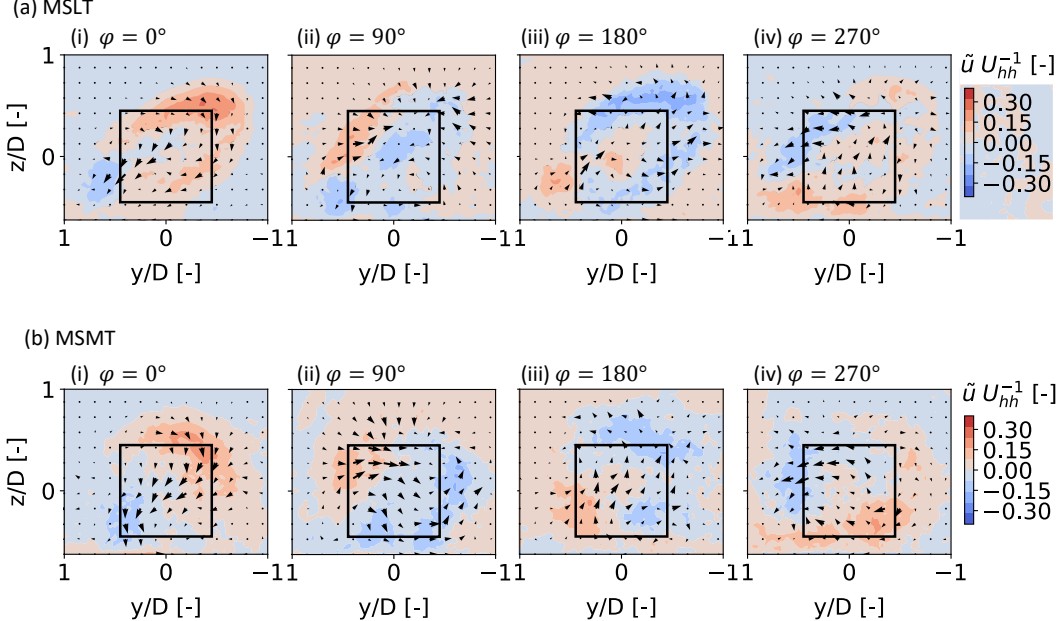

**Figure 18.** Cross sections of $\tilde{u}$ for the $n = -1$ case (ccw helix) at four phase angles for both the (a) MSLT and (b) MSMT conditions. The vectors are $\tilde{v}$ and $\tilde{w}$ and have the same scaling according to magnitude in all sub panels, and the flow is viewed from downstream looking upstream. Panel (a) is repeated from Fig. 15 for convenient side-by-side comparison.

## 4    Conclusions

This article is one of two in a companion paper series addressing the characteristics and performance of two wake-control strategies (i.e., WS and WM) based on measurement-backed, realistic offshore inflow conditions including those with low

turbulence and high veer that have not been a focus of previous research. Complementing the analyses on turbine quantities of interest in Frederik et al. (2025), this paper worked to elucidate the fluid-dynamic reasons for the observed changes in turbine performance by considering the wake behavior in single-turbine simulations. To these ends, we calculated the budget of MKE over two different control volumes of interest. The first control volume was square with the same cross-sectional area as a hypothetical downstream rotor and therefore helped pinpoint the source of recovery for turbines in a shallow array

that is exactly aligned with the wind direction. It was observed that the WS case derives nearly all its MKE benefit over the baseline from mean convection. The WM cases, on the other hand, benefited from increased turbulent entrainment, especially from coherent structures that produce noticeable phased-resolvable entrainment patterns. For a control volume with a wider spanwise dimension designed to study the sensitivity of the MKE recovery to small, stochastic variations in the wind direction, the WS case and the $n = 0$ case of WM were shown to be particularly effective at drawing down momentum from above. In

the case of WS, this is a consequence of increased mean flow down through the top-tip plane, which may be initiated by the CVP generated by the wake deflection. In the case of the WM $n = 0$ case, the improved MKE recovery stems from both a





strong burst of turbulent entrainment at $\varphi \approx 180°$ and from increased mean flow from above. The final analysis considered the phase-averaged results for a related inflow condition but one with higher turbulence and lower veer. The combined effect of the changes to turbulence and veer resulted in worse performance for the WS strategy but improved performance for the $n = -1$
WM strategy. A hypothesis derived from flow diagnostics is that both results relate to the removal of veer and the associated skewing of the wake. The importance of veer on wake-control strategies should not be understated, especially considering that the achieved levels of veer in this study were yet smaller than the measured values. There is thus need for more investigation of the physics of wake-control technology in offshore environments.

*Author contributions.* Sandia National Laboratories (KB, LC, GY, DH, ND) developed the precursor and turbine simulations used in this
article, as well as developing the control-volume analysis and writing the manuscript. The NREL team (JF, ES, PF) contributed to conceptualization, design of the test matrix, and writing.

*Competing interests.* At least one of the (co-)authors is a member of the editorial board of Wind Energy Science. The authors have no other competing interests to declare.

*Acknowledgements.* Sandia National Laboratories is a multimission laboratory managed and operated by National Technology & Engineer-
ing Solutions of Sandia, LLC, a wholly owned subsidiary of Honeywell International Inc., for the U.S. Department of Energy's National Nuclear Security Administration under contract DE-NA0003525.

This work was authored in part by the National Renewable Energy Laboratory, operated by Alliance for Sustainable Energy, LLC, for the U.S. Department of Energy (DOE) under Contract No. DE-AC36-08GO28308. This research used resources of the Oak Ridge Leadership Computing Facility at the Oak Ridge National Laboratory, which is supported by the Office of Science of the U.S. Department of Energy
under Contract No. DE-AC05-00OR22725.



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
