# Peer review of "Comparison of wind-farm control strategies under realistic offshore wind conditions: wake quantities of interest"

_Wind Energy Science, 2024_

## Referee Comment (RC2)

The current paper contains an interesting investigation of fluid dynamics mechanisms and mean kinetic energy fluxes behind the increased wake recovery observed in the most popular wake flow control mechanisms. The paper provides sufficient reference to literature, outlines the methodology in detail, and discusses the mean kinetic energy budget for both "aligned" and "varying wind direction" cases. Specific attention is further given to the fluxes through the top surface which are highly relevant for large and wake-saturated wind turbine arrays.

I believe the paper presents very interesting results and conclusions, and the majority of my comments below are not criticisms to the methodology and analysis (which I believe is sound), but rather to the presentation and discussion, as I believe some modifications could be made to further increase paper quality and readability and make the most important findings more explicit.

- The paper is rather long and, while this is not per se a problem in itself and final formatting will reduce the amount of pages, I believe the length of the paper impacts the readability. Certain parts could be made less verbose, shortened or moved to Appendix to improve the overall storyline of the paper. I leave it up to the authors and editors to decide, but some examples where I believe improvements could be made are

    o Section 2.3 on inflow conditions spans over 2 full pages of text + 2 tables and a figure, I believe the main idea of having representative inflow conditions are worthwhile including in the main text, but the details of, e.g., lidar filtering, could be moved to Appendix.

    o Section 3.2.1 on verification of technique could be moved to appendix without impacting the storyline of the paper

    o Figure 9 is a copy of Figure 8 for the "varying wind direction" case, and is not explicitly discussed by itself. I feel this figure could easily move to appendix or even be omitted.

    o 5 control techniques are discussed: WS, WM Pulse, WM Helix, WM side-to-side, and WM top-down. However, they are not investigated to the same amount of detail, and the reasons for this are a bit lost in the manuscript text. Especially side-to-side and top-down are only discussed very sparsely in the text, but included in most figures which makes them quite a lot busier and more difficult to analyse. I am wondering whether they actually contribute much to the analysis in the current manuscript, or could be omitted / moved to appendix after their introductions in Tables 7 and 8.

- Figure 2 shows the domain for the control volume analysis. The caption and manuscript highlight that the turbine is not part of this domain, but from looking at the figure, it does appear to be inside since the turbine footprint is in the blue shadow of the CV domain. I'd suggest revising the figure slightly.

- Section 2.2 details the LES setup. Although the description of the setup is generally satisfactory, for (stable) ABL simulations, initialization details matter, as significant unsteadiness in mean flow profiles can remain in improperly initialized simulations. However, these details are not elaborated or insufficiently detailed for reproducibility (e.g. What are the initial profiles of temperature and velocity, is a wind angle controller used to achieve a desired wind direction at hub height, the spinup period is detailed as "tens of thousands of seconds"). I would suggest to include these details in Appendix.

- Line 225 and Figure 4 show that the LES cannot reproduce the strong veer observed in the NY Bight lidar measurements. This is an interesting observation, also in light of the statement in the last line of the conclusion. Out of interest, do the authors have a hypothesis why the veer in the observations is so much stronger?

- Section 3.1 and specifically Figure 7 show that, after 10 D, WS achieves the strongest wake recovery which gives the impression that this is the most suitable technique for power maximization. However this does not give the complete story as the power loss in T1 is not included in the analysis. Although I understand the scope of the current paper is on the wake behavior rather than the achieved power gains, I believe a small note on the power losses in T1 could help put the comparison between techniques in better perspective.

- The titles of section 3.2.2 (Aligned wind direction) and especially 3.2.3 (Varying wind direction) in my opinion came across as confusing since the analysis is all performed on a constant wind direction LES. Perhaps they could be titled "narrow control volume" and "wider control volume" and the link to aligned vs. uncertain wind directions could be made in the manuscript.

- (Optional) I am wondering whether some of the figures would benefit from plotting the difference with the baseline rather than the absolute value, e.g. Figure 11 / 12. Upon reading the discussion in the text it takes the reader quite some work to identify the related features in these figures.

- Reporting of units and variables is highly inconsistent throughout the manuscript to the point where it becomes confusing and different conventions are used even within the same figure, please homogenize. Some examples

  - Around line 214: K-hr, K-m/s, ...

  - Table 3: m s^{ -1}

  - Figure 4: m / s

  - Figure 6: u U_{hh}^{-1}, x/D

---

## Author Comment (AC1)

April 9, 2025

We thank the reviewer for their helpful and constructive comments. The reviewer's comments and questions are addressed below, and we have made a number of changes to the manuscript text and figures. Manuscript text is given in *italics*, and new textual content is further formatted in **bold**.

**Reviewer 1**

The paper examines the mechanics of wind farm flow control strategies under realistic offshore conditions, specifically high veer. The analysis focuses on a budget analysis of mean kinetic energy, comparing five flow control strategies. This topic is highly relevant for the industrial application of wind farm flow control and this paper delivers some critical insight into the efficacy of different strategies under more complex inflow conditions. The paper's methodology is sound and well explained, and the analysis thoroughly founded on the presented data. I therefore recommend the paper for publication with minor revisions.

1. The methodology of the paper is very thorough, from the sophisticated determination of inflow conditions to generate to the validation of the budget analysis. However, I believe the thoroughness obstructs the readability of the article in some instances. Therefore, I recommend to move some parts into an appendix, in particular the majority of current section 2.3 and section 3.2.1.

This suggestion is well received, and we have moved four paragraphs and a table from Section 2.3 to an appendix titled: *Filtering and Processing of Floating Lidar Measurements*. This appendix now contains all the finer details about the processing of measurement data that might be relevant for reproducibility but not relevant to the general narrative of the article. The text remaining in Section 2.3 is related only to the comparison of the simulated results with the measured ones, and we believe this comparison is an important aspect that deserves to remain in the main body of text due to the sensitivity of the performance of wake-control strategies to the ambient flow conditions.

Related to Section 3.2.1, we would prefer to leave this section on verification of the MKE technique in its current position to emphasize to readers the precision to which the MKE budget has been calculated, considering how important the MKE budget is to this paper's conclusions. We hope the reviewer agrees after reviewing all the changes to the manuscript.

2. In section 2.2, I think the used model could be described in some more detail, especially how OpenFAST was set up and how the rotational speed of the turbine is controlled.

The authors agree that the original content related to the OpenFAST and ROSCO setup was not thoroughly discussed. We have made two changes to address this. First, a new paragraph has been added in Section 2.5 regarding how the rotational speed of the turbine was controlled:

While WS and AWM alter the yaw and blade-pitch control, respectively, neither strategy alters the baseline generator-torque control, which tracks optimum tip-speed ratio. During WS, the controller thus reduces generator torque and rotor speed as the turbine yaws away from the oncoming wind since the component of inflow velocity normal to the rotor face is reduced. During AWM, the controller accommodates reductions in rotor loading by lowering the torque and vice-versa, given some phase lag. The pulse case sees significant fluctuations of torque due to the collective pitch control, but the other, individual-pitch AWM strategies have relatively constant torque.

Second, in Section 2.2 where the OpenFAST simulations are introduced, we have added a pointer to specific details of note that are discussed in the companion article, which was chosen as the article to describe the turbine setup in greater detail:

Specific settings for the turbine in this study are provided in the companion article Frederik et al. [2025] including details related to the time step and solvers used.

3. The authors write that the ratio of smearing width to cell size is 0.8, this appears very low to me and well below commonly found recommendations. If it is indeed not a misprint the authors should add some justification or point to validation.

The reviewer is correct that the value of  $\varepsilon/\Delta x = 0.8$  is considerably lower than the value of 2 or so often recommended. This is not a misprint but the result of a study to set an appropriate  $\varepsilon/\Delta x$ . This calibration process saw  $\varepsilon$  simulated with values of 1-5 as well as 10, and the results were compared to the power curve of the IEA 15 MW turbine run in OpenFAST alone. A value of  $\varepsilon = 2$  produced excellent agreement with this reference and was adopted for this work, thus producing the aforementioned  $\varepsilon/\Delta x$  given the most refined grid size around the turbine of  $\Delta x = 2.5m$ .

This study is documented online at:

https://exawind.github.io/amr-wind/walkthrough/calibration.html

and this reference has been added to the manuscript, along with some additional words as described below:

The ALM is defined with an isotropic Gaussian projection function with spreading parameter  $\varepsilon/\Delta x = 0.8$ , and this value, though low compared to some recommendations, was determined based on agreement of the LES power curve with the OpenFAST one as described in Yalla [2024].

4. I find the double naming convention for the wake mixing strategies, i.e. giving the modal numbers and the name, applied throughout the paper too long, it would improve readability to use either the modal numbers or the names.

We have followed this suggestion and changed all references to just a single naming convention (save for the first mention where both naming conventions are introduced). The single naming convention chosen is the more intuitive "pulse", "side-to-side", etc., which conforms with the naming convention used in the companion article.

5. In the companion paper, the authors speak of active wake mixing instead of wake mixing, for a reader that reads both papers it would be much easier if the same nomenclature is used.

The authors have implemented this suggestion to conform with the companion paper by changing all references of wake mixing (WM) to active wake mixing (AWM).

- 6. Furthermore, I have the following minor suggestions:
  - section 2.1: I dont think it is mentioned anywhere explicitly that the origin is at the hub of the turbine, please add this information for the reader's convenience.

This information has been added to the caption of Figure 2:

The x origin of the coordinate system is at the tower centerline, and the rotor...

• I. 133: why is it called AP?

This terminology was borrowed from Van der Hoek et al. [2024], who used the term  $f_{AP}$  to describe the fraction of available power for a hypothetical turbine in the wake compared to one in the free stream inflow. While we recognize that AP is not a standard term in literature (yet!), we adopted it since a simple P could be misconstrued as the turbine electrical power in the expression given in Table 4:  $AP_{out}C_p$ .

• tables 5,6,7: there is a lot of white space in the tables that can be removed.

We agree that there is a lot of unnecessary white space. Based on previous publishing experience with WES, it is the authors' understanding that such formatting adjustments will be kindly made by the typesetters of the article and need not be bothered by the authors. If that is *not* the case, we will be glad to make a further revision with improved formatting.

• figure 14: the color scheme makes it quite difficult to subplot a, especially in greyscale. Although a small change, we have increased the contrast on the color bars in Figure 14 as well as in Figures 8, 9, 15, and 17, which all use the same color scheme in the bar chart format. In Figures 8 and 9, where the color scheme is first introduced, we preferred to use four similar colors to represent the turbulent entrainment from the four side surfaces of the control volume. This makes it easier to visually ascertain the overall contribution from different types of terms (i.e., mean convection, turbulent entrainment, etc.). Since Figures 14, 15, and 17 are derivatives of the analysis in Figures 8 and 9, we wanted to retain the same color scheme, and thus the same adjustment has been made to five figures. We believe that the small adjustment to the orange color contrast will make the figure easier to interpret, even in greyscale. • figure 14 and 15: the caption says the panels are shown looking upstream, I think it should read downstream.

This error has been corrected; thank you.

- Figure 17 is very large, the size can be reduced significantly This change has been implemented.
- Throughout the figures the authors use an inconsistent formatting for the normalizations, in most figures (e.g. figure 5), the axis labels are given as (coordinate) / (normalization scale), however, in some figures, e.g. figure 7, the x-axis label reads xD-1. I think a consistent style would be nice.

This inconsistency has been fixed in both figures and text as we now use only the  $xD^{-1}$  form to denote unit quantities in the denominator. This edit was made in  $\approx 25$  places throughout the text; thank you for pointing it out.

April 9, 2025

We thank the reviewer for their helpful and constructive comments. The reviewer's comments and questions are addressed below, and we have made a number of changes to the manuscript text and figures. Manuscript text is given in *italics*, and new textual content is further formatted in **bold**.

**Reviewer 2**

The current paper contains an interesting investigation of fluid dynamics mechanisms and mean kinetic energy fluxes behind the increased wake recovery observed in the most popular wake flow control mechanisms. The paper provides sufficient reference to literature, outlines the methodology in detail, and discusses the mean kinetic energy budget for both "aligned" and "varying wind direction" cases. Specific attention is further given to the fluxes through the top surface which are highly relevant for large and wake-saturated wind turbine arrays.

I believe the paper presents very interesting results and conclusions, and the majority of my comments below are not criticisms to the methodology and analysis (which I believe is sound), but rather to the presentation and discussion, as I believe some modifications could be made to further increase paper quality and readability and make the most important findings more explicit.

- The paper is rather long and, while this is not per se a problem in itself and final formatting will reduce the amount of pages, I believe the length of the paper impacts the readability. Certain parts could be made less verbose, shortened or moved to Appendix to improve the overall storyline of the paper. I leave it up to the authors and editors to decide, but some examples where I believe improvements could be made are
  - Section 2.3 on inflow conditions spans over 2 full pages of text + 2 tables and a figure, I believe the main idea of having representative inflow conditions are worthwhile including in the main text, but the details of, e.g., lidar filtering, could be moved to Appendix.

This suggestion is well received, and we have moved four paragraphs and a table from Section 2.3 to an appendix titled: *Filtering and Processing of Floating Lidar Measurements*. This appendix now contains all the finer details about the processing of measurement data that might be relevant for reproducibility and context but not as relevant to the general narrative of the article. The text remaining in Section 2.3 is related only to the comparison of the simulated results with the measured ones, and we believe this comparison is an important aspect that deserves to remain in the main body of text due to the sensitivity of the performance of wake-control strategies to the ambient flow conditions.

 Section 3.2.1 on verification of technique could be moved to appendix without impacting the storyline of the paper

We would prefer to leave this section on verification of the MKE technique in its current position to emphasize to readers the precision to which the MKE budget has been calculated, considering how important the MKE budget is to this paper's conclusions. We hope the reviewer agrees after reviewing all the changes to the manuscript.

• Figure 9 is a copy of Figure 8 for the "varying wind direction" case, and is not explicitly discussed by itself. I feel this figure could easily move to appendix or even be omitted. We would like to keep this figure in the main body of the manuscript because it may have value to highlight one aspect of the wake performance that its corresponding tabular-format data does not: streamwise evolution of MKE recovery. Admittedly, this point was not captured in the original manuscript. We have added a small but significant piece of text to bring this point to light:

... Therefore, there is increased opportunity for turbulent MKE recovery from aloft through the top face as discussed in Sect. 1 related to the literature on larger-array wind farms and demonstrated here by the prolonged gains of  $\partial \phi_{turb,tp}/\partial x$ in Figure 9 versus Figure 8...

 5 control techniques are discussed: WS, WM Pulse, WM Helix, WM side-to-side, and WM top-down. However, they are not investigated to the same amount of detail, and the reasons for this are a bit lost in the manuscript text. Especially side-to-side and top-down are only discussed very sparsely in the text, but included in most figures which makes them quite a lot busier and more difficult to analyze. I am wondering whether they actually contribute much to the analysis in the current manuscript, or could be omitted / moved to appendix after their introductions in Tables 7 and 8.

We agree that the narrative does not give equal weight to the discussion of the four AWM methods nor any explanation as to why this is so. To amend this, we have added language to transition from the overview of all the cases earlier in the article into the final analyses focusing only on the fewer, stronger-performing cases. In doing so, we have added a new section to help punctuate this change. This section begins as:

**3.4 Phase-averaged analysis of entrainment**

In light of this unsteady, periodic nature of the AWM strategies, a phase-averaged analysis of the flow fields is useful to pinpoint the source of the increases in  $-\overline{uu'w'}$ observed in Fig. 13. We focus the phase-averaged analysis (and other subsequent AWM analyses) on the pulse and ccw helix cases alone since these demonstrated high turbulent entrainment through the top surface in Fig. 13 and indeed demonstrated stronger performance over the side-to-side and up-and-down cases for  $x_{end}D^{-1}$  between 3 and 6 in Fig. 7...

2. Figure 2 shows the domain for the control volume analysis. The caption and manuscript highlight that the turbine is not part of this domain, but from looking at the figure, it does appear to be inside since the turbine footprint is in the blue shadow of the CV domain. I'd suggest revising the figure slightly.

Thank you for this comment; we have modified the figure to eliminate a spurious 0.15D offset in the rendering of the turbine's x-position. Further, we have increased the xy extent of the rendering of the ground surface.

3. Section 2.2 details the LES setup. Although the description of the setup is generally satisfactory, for (stable) ABL simulations, initialization details matter, as significant unsteadiness in mean flow profiles can remain in improperly initialized simulations. However, these details are not elaborated or insufficiently detailed for reproducibility (e.g. What are the initial profiles of temperature and velocity, is a wind angle controller used to achieve a desired wind direction at hub height, the spinup period is detailed as "tens of thousands of seconds"). I would suggest to include these details in Appendix.

We agree that these details were omitted and should be included. The text now includes:

The initial temperature profile was neutral for the first 500 meters of the ABL, followed by a 100 m inversion layer and then a gradual, 0.002  $K m^{-1}$  rise to the top of the domain. The initial profiles of wind speed and direction were uniform based on the specified values at hub height, and these were enforced during the simulation by actively adjusting the pressure gradient until the horizontally averaged wind velocity matched the desired values at hub height. On top of these initial profiles, small velocity and temperature perturbations were added near the surface to accelerate turbulence development, and the precursors were run for tens of thousands of seconds to establish fully developed turbulent flow including >25,000 s and >40,000 s, respectively, for the stable and near-neutral ABL conditions (to be described below) before collecting averaged statistics for the ABL.

4. Line 225 and Figure 4 show that the LES cannot reproduce the strong veer observed in the NY Bight lidar measurements. This is an interesting observation, also in light of the statement in the last line of the conclusion. Out of interest, do the authors have a hypothesis why the veer in the observations is so much stronger?

The inability of our simulation to achieve the larger veer values observed in measurements is due to the forcing strategy used to drive the ABL towards the target statistics, which limits the number of metrics which can be matched simultaneously. The current ABL forcing strategy iteratively adjusts the surface roughness and surface heating or cooling rate until specific ABL metrics are satisfied. In this study, the hub-height turbulence intensity and rotor-averaged shear exponent were chosen as the primary metrics to match, meaning that some differences in veer were allowed to develop (the horizontally-averaged hub-height wind speed and direction were guaranteed to match given the pressure gradient forcing described above). We acknowledge that this forcing does not replicate the exact conditions experienced in offshore measurements, as the actual wind speed/direction and surface heating/cooling vary in time. In the absence of a wind-angle controller, a larger, more realistic domain (i.e., a meso-micro-scale forcing strategy) may be necessary to increase veer.

5. Section 3.1 and specifically Figure 7 show that, after 10 D, WS achieves the strongest wake recovery which gives the impression that this is the most suitable technique for power maximization. However this does not give the complete story as the power loss in T1 is not included in the analysis. Although I understand the scope of the current paper is on the wake behavior rather than the achieved power gains, I believe a small note on the power losses in T1 could help put the comparison between techniques in better perspective.

This is a helpful comment, and we have made the following additions to second paragraph of Section 3.1:

A more quantitative perspective on the wake recovery is afforded by Figure 7, which plots the streamwise development of MKE recovery using the near-wake-subtracted and normalized  $AP_{out}$ . Near-wake subtraction was performed to emphasize differences in the rates of recovery between cases, however, it should be noted that neither the near-wake-relative wake recovery depicted in Figure 7 nor the absolute wake deficit (not shown) is the same as overall farm-level benefit since we have not accounted for power losses on the upstream turbine, which are typically larger for WS than for WM as demonstrated in Frederik et al. [2025].

6. The titles of section 3.2.2 (Aligned wind direction) and especially 3.2.3 (Varying wind direction) in my opinion came across as confusing since the analysis is all performed on a constant wind direction LES. Perhaps they could be titled "narrow control volume" and "wider control volume" and the link to aligned vs. uncertain wind directions could be made in the manuscript.

We like this suggestion and have implemented it, changing the titles of both sections to the reviewer's suggestions. The first sentence of each section makes the link between the control volume width and the (un)certainty of the wind direction.

7. (Optional) I am wondering whether some of the figures would benefit from plotting the difference with the baseline rather than the absolute value, e.g. Figure 11 / 12. Upon reading the discussion in the text it takes the reader quite some work to identify the related features in these figures.

This point is well taken, and the authors have also had internal discussion about whether these figures should be plotted as absolute values or baseline-subtracted values. Given the presence of strong veer, which is relatively unusual in the wake-control literature as yet, we thought it was important to show the absolute values so that the reader fully understands the underlying baseline flow field. We are open to plotting baseline-subtracted values, as well, but do not want to extend the length of the paper unnecessarily.

- 8. Reporting of units and variables is highly inconsistent throughout the manuscript to the point where it becomes confusing and different conventions are used even within the same figure, please homogenize. Some examples
  - Around line 214:  $K hr, K m/s, \dots$
  - Table 3:  $ms^{-1}$
  - m/s
  - Figure 6:  $uU_{hh}^{-1}, x/D$

This inconsistency has been fixed in both figures and text as we now use only the  $xD^{-1}$  form to denote unit quantities in the denominator. This edit was made in  $\approx 25$  places throughout the text; thank you for pointing it out.

**References**

- Joeri Frederik, Eric Simley, Kenneth Brown, Gopal Yalla, Lawrence Cheung, and Paul Fleming. Comparison of wind farm control strategies under a range of realistic wind conditions: turbine quantities of interest. Wind Energy Science, 2025. In review.
- Daan Van der Hoek, Bert Van den Abbeele, Carlos Simao Ferreira, and Jan-Willem van Wingerden. Maximizing wind farm power output with the helix approach: Experimental validation and wake analysis using tomographic particle image velocimetry. Wind Energy, 27(5):463–482, 2024.
- Gopal Yalla. Actuator line model calibration. https://exawind.github.io/amr-wind/ walkthrough/calibration.html, 2024. Accessed: 2024-10-30.

---

## Referee Report (RR1)

**Review**

I believe the changes made by the authors have further improved the manuscript and it is now ready for publication.

I only have minor editorial comments:
l. 378: "..from $xD^{-1} = 1.5D$" remove one of the Ds
section 4.3.1: I found the varying signs of $u^2w$ in this section somewhat hard to track, I think it would be easier to stick to a single sign.
l. 431: "... observed in Fig. 11(b)", I think you mean fig 11(a).
l. 431: "...enhancement of the region of positive $u^2w$", I believe it should be $-u^2w$